# Arithmetic Without Algorithms: Language Models Solve Math with a Bag of Heuristics

**Yaniv Nikankin**[1][*]  **Anja Reusch**[1]  **Aaron Mueller**[1,2]  **Yonatan Belinkov**[1]
[1]Technion – Israel Institute of Technology   [2]Northeastern University

## Abstract

Do large language models (LLMs) solve reasoning tasks by learning robust generalizable algorithms, or do they memorize training data? To investigate this question, we use arithmetic reasoning as a representative task. Using causal analysis, we identify a subset of the model (a circuit) that explains most of the model's behavior for basic arithmetic logic and examine its functionality. By zooming in on the level of individual circuit neurons, we discover a sparse set of important neurons that implement simple heuristics. Each heuristic identifies a numerical input pattern and outputs corresponding answers. We hypothesize that the combination of these heuristic neurons is the mechanism used to produce correct arithmetic answers. To test this, we categorize each neuron into several heuristic types—such as neurons that activate when an operand falls within a certain range—and find that the unordered combination of these heuristic types is the mechanism that explains most of the model's accuracy on arithmetic prompts. Finally, we demonstrate that this mechanism appears as the main source of arithmetic accuracy early in training. Overall, our experimental results across several LLMs show that LLMs perform arithmetic using neither robust algorithms nor memorization; rather, they rely on a "*bag of heuristics*". [1]

## 1 Introduction

Do large language models (LLMs) implement robust reusable algorithms to solve tasks, or are they merely memorizing aspects of the training distribution? This distinction is crucial (Tänzer et al., 2022; Henighan et al., 2023): while memorization might suffice for limited problem sets, true algorithmic comprehension allows for generalization and efficient scaling to new problems.

Arithmetic reasoning provides a lens for this investigation, as it can be solved using various methods: learning known algorithms, developing novel approaches, or by memorizing vast quantities of input-output pairs. Thus, we ask the following: Do LLMs implement robust algorithms to correctly complete arithmetic prompts, similar to children learning vertical addition to add two numbers, or do LLMs merely memorize the arithmetic prompts that appear in their vast training data?

Previous studies have made progress in identifying arithmetic mechanisms in LLMs. Stolfo et al. (2023) and Zhang et al. (2024) have identified a subset of model components (*a circuit*) responsible for arithmetic calculations in several LLMs and characterized the information flow between them. Zhou et al. (2024) suggested that pre-trained LLMs use features in Fourier space to accurately answer addition prompts. However, Stolfo et al. (2023) and Zhang et al. (2024) stopped short of elucidating the mechanism implemented by the circuit they identified—a required feat to understand the trade-off between generalization and memorization in this task. Zhou et al. (2024) studied addition prompts in models that were *fine-tuned* on arithmetic data. Their findings regarding Fourier features are significant, but we claim these represent only a part of a more complex mechanism. Our work aims to bridge these gaps: we investigate how the arithmetic circuit works qualitatively—and specifically, whether it implements a mathematical algorithm or memorizes arithmetic training data.

To do so, we reverse-engineer the arithmetic mechanism applied by LLMs. We use causal analysis to examine their arithmetic circuits, focusing on individual neurons within the circuit responsible

---

[*]Correspondence to: `yaniv.n@cs.technion.ac.il`
[1]Code at `https://github.com/technion-cs-nlp/llm-arithmetic-heuristics`

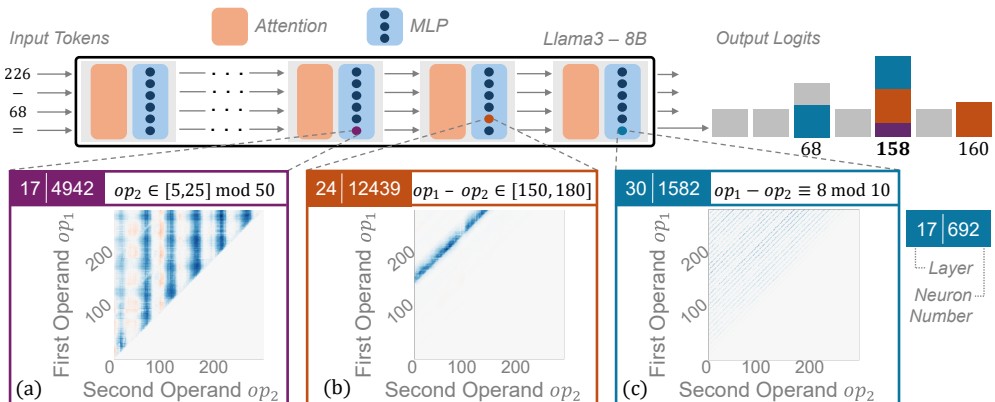

Figure 1: **Bag of heuristics visualization.** We show that transformer LLMs solve arithmetic prompts by combining several unrelated heuristics, each activating according to rules based on the input values of operands, and boosting the logits of corresponding result tokens. These heuristics are manifested in single MLP neurons in mid to late layers.

for generating the correct answer. Our analysis reveals that a sparse subset of neurons is sufficient for accurate responses, with each neuron implementing a distinct *heuristic*. Each heuristic fires for a specific pattern in input operands or in their combination, and some increase the logits of relevant result tokens accordingly. For instance, one heuristic (Figure 1b) increases logits of tokens between 150 and 180 for subtraction prompts whose answer falls within this range. By examining these neurons, we classify each into one or more heuristic types. For example, the mentioned neuron (Figure 1b) falls within the type of result-range heuristics, which promote continuous value ranges. We discover that successful prompt completion relies on a combination of several unrelated heuristic types, forming a *"bag of heuristics"* approach. This finding suggests that LLMs may not be employing a single, cohesive algorithm for arithmetic reasoning, nor are they memorizing all possible inputs and outputs; rather, they deploy a collection of simpler rules and patterns.

We investigate if the bag of heuristics emerges as the primary arithmetic mechanism from the onset of training, or whether it overrides an earlier mechanism. To do that, we analyze how the heuristics evolve over the course of training. We show arithmetic heuristics appear throughout the model training, gradually converging towards the heuristics observed in the final checkpoint. Furthermore, we provide evidence that the bag of heuristics mechanism explains most of the model's behavior even in early stages, indicating it is the main mechanism used for solving arithmetic prompts.

We contribute by providing a high-resolution understanding of the mechanism that LLMs use to answer arithmetic prompts. We (i) show pre-trained LLMs implement a "bag of heuristics" approach, (ii) investigate when and why this mechanism fails to generalize, and (iii) discover how it emerges across training. This allows us to better understand the source of current capabilities and limitations of LLMs in arithmetic reasoning—a finding that could apply to additional reasoning tasks.

## 2 ARITHMETIC CIRCUIT DISCOVERY

In transformer-based LLMs, a **circuit** (Elhage et al., 2021) refers to a minimal subset of interconnected model components (multi-layer perceptrons (MLP) or attention heads) that perform the computations required for a specific task. We locate, and later analyze, the circuit responsible for arithmetic calculations.

### 2.1 CIRCUIT DISCOVERY AND EVALUATION

**Models and Data** We analyze four LLMs: Llama3-8B/70B (Dubey et al., 2024), Pythia-6.9B (Biderman et al., 2023), and GPT-J (Wang & Komatsuzaki, 2021). For each, we locate and analyze the circuit responsible for arithmetic calculations. We focus on Llama3-8B in the main paper and report similar results for the additional models in Appendix I. We use pre-trained models without fine-tuning them on arithmetic prompts, as our goal is to uncover the mechanisms induced by typical

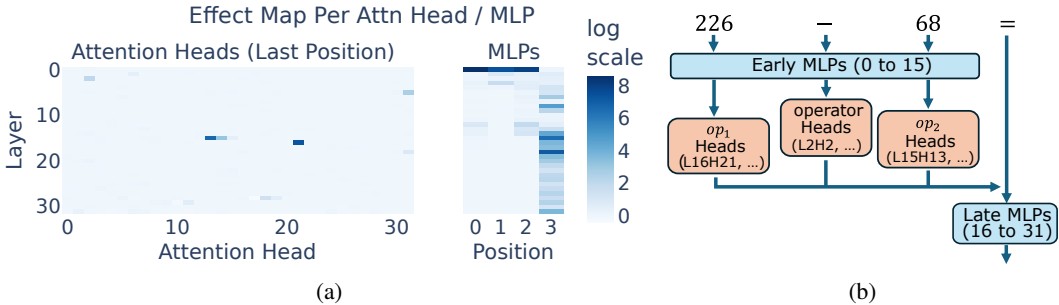

(a)                                                                                    (b)

Figure 2: **Llama3-8B arithmetic circuit discovery results. (a):** Few attention heads have a high effect on arithmetic prompts. Most MLPs take part in the computation. The first MLP noticeably affects operand and operator positions, while mid- and late-layer MLPs influence the final position. **(b):** The arithmetic circuit in Llama3-8B. The attention heads project token information to the last position, where the middle- and late-layer MLPs promote the logits for the correct answer.

language model training. Each model tokenizes positive numbers as a single token, up to some limit; e.g., in Llama3-8B, numbers in $[0, 1000]$ are tokenized to a single token. To locate the arithmetic circuit, we use two-operand arithmetic prompts with Arabic numerals and the four basic operators $(+, -, \times, \div)$, such that each prompt has four tokens: $op1$, the operator, $op2$, and the "$=$" sign. We sample a list of 100 prompts per operator for circuit discovery, and an identical amount for evaluation. Each prompt is chosen so that both its operands and result will be tokenized to a single token; e.g., in Llama3-8B, the operands and result must be between 0 and $1,000$. Unlike previous studies (Stolfo et al., 2023), we do not use in-context prompting, to ensure the circuit does not include any component not directly linked to arithmetic calculations. To reduce noise and ensure the circuits only contain components responsible for correct arithmetic completions, we only use prompts that are correctly completed by the model, similar to previous studies (Wang et al., 2022; Prakash et al., 2024). Throughout the paper, the prompt "$226 - 68 =$" is used as a running example.

**Method**   To locate the circuit components, we conduct a series of activation patching experiments (Vig et al., 2020) that allow us to assess the importance of each MLP and attention head at each sequence position. Each experiment involves sampling a prompt $p$ with result $r$ from the dataset (for example, "$226 - 68 =$"), and sampling a random counterfactual prompt $p'$ that leads to a different result $r'$ (for example, "$21 + 17 =$"). After pre-computing the activations of the model for the counterfactual prompt $p'$, we introduce the prompt $p$ to the model. We intervene on (*"patch"*) the computation, which means we replace the activation of a single MLP layer or attention head with its pre-computed activation for $p'$. Following Stolfo et al. (2023), we observe how this intervention affects the probabilities of both answer tokens, $r$ and $r'$, by measuring the following:

$$\mathrm{E}(\mathrm{r}, \mathrm{r}') = \frac{1}{2} \left[ \frac{\mathbb{P}^*(r') - \mathbb{P}(r')}{\mathbb{P}(r')} + \frac{\mathbb{P}(r) - \mathbb{P}^*(r)}{\mathbb{P}^*(r)} \right] \tag{1}$$

where $\mathbb{P}$ and $\mathbb{P}^*$ are the pre- and post-intervention probability distributions, respectively. The two summands in Equation (1) increase if patching raises the probability of $r'$ or decreases the probability of $r$, respectively. High effect for an intervention on a component indicates its high importance in prompt calculation. The effect is averaged across prompts and measured separately per component.

**Results**   The patching results, shown in Figure 2a, reveal that the MLP layers affect the output probabilities more than the attention heads. The first MLP affects the representation at the operator and operand positions (see Appendix B.1), while middle- and late-layer MLPs exhibit a strong effect at the final position, likely reflecting their role in predicting the answer token in that position (further discussed in Section 2.2). Figure 2a also shows that very few attention heads are important to the circuit. Each such attention head copies information from a single position (either an operand or operator position) to the final position (see Appendix B.2). Figure 2b summarizes the information flow within the circuit, consistent in structure with prior work (Stolfo et al., 2023).

To evaluate the circuit **c**, we measure its faithfulness (Wang et al., 2022), the proportion of the full model's behavior on arithmetic prompts that can be explained solely by the circuit. To measure

faithfulness, we first pre-compute mean activations for each model component (in each position) across *all* arithmetic prompts. We then intervene on the evaluation prompts by replacing non-circuit component activations with their means. To quantify performance, we measure $\text{NL}(\mathbf{c})$, the logit of the correct answer token normalized by the maximal logit, as a proxy for accuracy, when mean-ablating all components not in the circuit $\mathbf{c}$. The circuit's faithfulness is calculated as:

$$\text{F}(\mathbf{c}) = \frac{\text{NL}(\mathbf{c}) - \text{NL}(\emptyset)}{\text{NL}(\mathbf{M}) - \text{NL}(\emptyset)} \tag{2}$$

where $\mathbf{M}$ is the entire model and $\text{NL}(\mathbf{M})$ is the normalized correct-answer logit when no component is ablated (always 1.0 for correctly completed prompts). $\text{NL}(\emptyset)$ is the normalized correct answer logit when all components are mean-ablated. This formula normalizes faithfulness to a $[0.0, 1.0]$ range.

The circuit achieves a high faithfulness of 0.96 on average across the four arithmetic operators; i.e., the circuit accounts for 96% of the entire model's performance. We can therefore conclude that the components identified in this section comprise the arithmetic circuit, and explain most of the model's accuracy for arithmetic prompts. See Appendix A for results across various circuit sizes, and a discussion of Pareto-optimality with respect to faithfulness and size.

## 2.2 IDENTIFYING ANSWER-PROMOTING COMPONENTS

To understand the mechanism implemented by the circuit to promote the correct answer, we first search for the specific circuit components that increase the probability of the correct answer. For this, we employ linear probing (Belinkov, 2022). For each layer $l$ and sequence position $p$, we train a linear classifier $f_{l,p} : \mathbb{R}^d \to \mathbb{R}^{1000}$ (where $d$ is the dimension of each layer's output representation) using a training set of correctly completed arithmetic prompts. We pass these prompts through the model and calculate the output representation $h^{l,p} \in \mathbb{R}^d$ at layer $l$ and position $p$ for

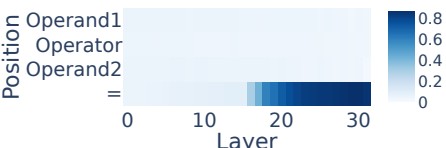

Figure 3: **Answer token probe accuracy.** The linear probes are successful in extracting the correct answer from the final position, starting at layer 16.

each prompt. The classifier $f_{l,p}$ receives $h^{l,p}$ as input, and outputs a probability distribution over the $1,000$ possible arithmetic answers. The classifier $f_{l,p}$ is evaluated on a separate test set of correctly completed prompts, showing to what extent the correct answer can be extracted from the output representation at layer $l$ and position $p$.

We find that the answer can only be extracted with high accuracy from the final position (Figure 3), after the token representation is processed by the later layers of the model, starting from layer $l = 16$. Given that the arithmetic circuit contains only MLPs in these layers (Figure 2), this suggests that these MLPs in layers $[16, 32]$ are the components that write the correct answer into the representation at the last position. The following section zooms into these middle- and late-layer MLPs, and presents evidence for the role they play in generating the correct answer— specifically, in how they promote the correct answer token through a combination of many independent arithmetic heuristics.

# 3 MLP NEURONS IMPLEMENT ARITHMETIC HEURISTICS

## 3.1 DECOMPOSING CIRCUIT MLPS TO INDIVIDUAL NEURONS

Having shown that the model generates the arithmetic answer in middle- and late-layer MLPs at the final position $p = 4$, we zoom in on these MLPs and their calculations at this position to investigate the implemented mechanism. The MLP at layer $l$ can be described by the following equation:

$$\mathbf{h}^l_{out} = \text{MLP}_{in}\left(\mathbf{h}^l_{in}\right) \cdot \mathbf{W}^l_{out} = \mathbf{h}^l_{post} \cdot \mathbf{W}^l_{out} = \sum_{n=0}^{d_{mlp}} \mathbf{h}^{l,n}_{post} \mathbf{v}^{l,n}_{out} \tag{3}$$

where $\mathbf{h}^l_{in}, \mathbf{h}^l_{out} \in \mathbb{R}^d$ are the input and output representations of the MLP at layer $l$, respectively. $\mathbf{h}^l_{post} \in \mathbb{R}^{d_{mlp}}$ is the output of the up-projection of the MLP, [2] where we define the $n^{\text{th}}$ value $\mathbf{h}^{l,n}_{post} \in$

---

[2]The up-projection $\text{MLP}_{in}$ is implemented differently in each LLM we analyze. See Appendix C.

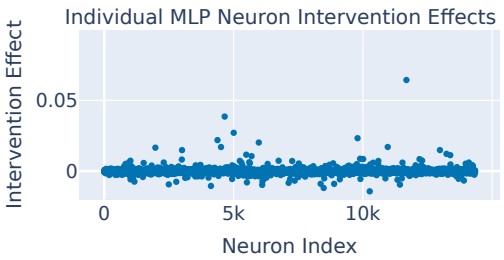

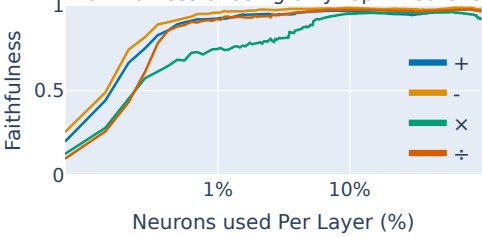

(a) We intervene on each neuron in MLP layer $l = 17$. Few neurons in this and other middle- and late-layer MLPs have a high effect on arithmetic prompts.

(b) We measure the faithfulness when including only a fraction of high-effect neurons in the circuit. This circuit achieves high faithfulness.

Figure 4: Analyzing effect of individual circuit MLP neurons. Our results demonstrate that a small amount of neurons is required to correctly predict the result.

$\mathbb{R}$ as a neuron. $\mathbf{W}_{out}^l \in \mathbb{R}^{d_{mlp} \times d}$ is the output projection matrix, and $\mathbf{v}_{out}^{l,n}$ is its $n^{\text{th}}$ row vector. Biases are omitted. By expressing the output representation $\mathbf{h}_{out}^l$ as a linear combination of row vectors $\mathbf{v}_{out}^{l,n}$ and their corresponding neuron activations $\mathbf{h}_{post}^{l,n}$, we can identify the neurons that most affect the completion of arithmetic prompts.

To measure the effect of each neuron $\mathbf{h}_{post}^{l,n}$, we perform activation patching experiments on individual neurons (as described in Section 2.1), and measure the average effect across prompts. We observe that very few neurons have a high effect; an example for layer $l = 17$ is shown in Figure 4a. Additionally, we notice the neurons with the highest effect are different between operators. In fact, roughly 45% of the important neurons for each operator are unique (Appendix D). Thus, when analyzing the circuit at the neuron level, we analyze it as 4 separate circuits—one for each arithmetic operator. We hypothesize that for each operator, the highest-effect neurons are sufficient to explain most of the model's arithmetic behavior. To verify this, we measure the faithfulness of the arithmetic circuit when mean ablating non-circuit components and lower effect MLP neurons in middle- and late-layer MLPs. The results (Figure 4b) confirm that **only 200 neurons (roughly 1.5%) per layer are needed to achieve high faithfulness and correctly compute arithmetic prompts**.

## 3.2 MLP NEURONS ACT AS MEMORIZED HEURISTICS

To understand how the top-200 middle- and late-layer important MLP neurons contribute to the generation of correct answers, we view them as key-value memories (Geva et al., 2021).[3] In this view, the input to each MLP layer $\mathbf{h}_{in}^l$ is multiplied by a *key* (a row vector in the MLP input weight matrix) to generate a neuron activation $\mathbf{h}_{post}^{l,n}$ that determines how strongly does a *value* (a row vector $\mathbf{v}_{out}^{l,n}$ in the MLP output weight matrix) gets written to the MLP output (Equation (3)). Geva et al. (2021) demonstrated that keys correspond to specific topics or n-grams, triggering high neuron activations when these are given as input, and their corresponding values represent tokens that can serve as appropriate completions for these topics or n-grams. Building on this insight, we hypothesize that (i) in arithmetic contexts, keys correspond to numerical patterns, e.g., a neuron might activate strongly when both operands in an arithmetic operation are odd numbers; and (ii) the associated value vectors encode numerical tokens that represent plausible answers to the key patterns.

To test the first hypothesis, we investigate the activation pattern of the top-200 neurons in each layer. For each neuron $l, n$, we plot the activations $\mathbf{h}_{post}^{l,n}$ (at position $p = 4$) as a function of operand values, separately for each operator. We find that many neurons in the arithmetic circuit exhibit distinct, human-identifiable patterns. For instance, in "$226 - 68 =$", neuron $\mathbf{h}_{post}^{24,12439}$ shows high activation values for subtraction prompts with results between 150 and 180 (Figure 1). Additional examples are provided in Appendix J.

To verify the second hypothesis, we check whether the tokens embedded in the value vectors of the top neurons relate to their activation patterns. Using the Logit Lens (nostalgebraist, 2020), a method of projecting a vector $\mathbf{v} \in \mathbb{R}^d$ onto a probability distribution over the vocabulary space $\mathbb{P}^{d_{vocab}}$, we project each value vector $\mathbf{v}_{out}^{l,n}$ to find the numerical tokens whose logits are highest. This reveals

---

[3]Not to be confused with the attention heads' keys and values.

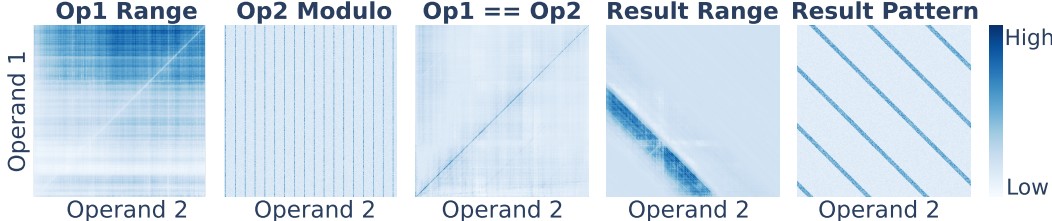

Figure 5: **Heuristic pattern examples.** Each heatmap is the activation pattern of an example neuron, implementing a specific **heuristic type**. Within the heatmap, each pixel at location $(op_1, op_2)$ represents the activation strength of the neuron under the addition prompt "$op_1 + op_2 =$".

two distinct patterns: First, in some neurons, the activation pattern depends on both operands *and* the value vector encodes the expected result of the arithmetic calculation (Figure 1b,c). We term such neurons *direct heuristics*. Second, in neurons where the activation pattern depends on a single operand, the value vectors often encode features for downstream processing, rather than the result tokens directly (Figure 1a). We term such neurons *indirect heuristics*. Next, we demonstrate how these heuristic neurons combine to produce correct arithmetic answers.

## 4 ARITHMETIC PROMPTS ARE ANSWERED WITH A BAG OF HEURISTICS

Observing the example prompt, "$226 - 68 =$", we have shown that it satisfies the pattern of several heuristic neurons, where each such neuron slightly increases the logit of the result token, $r = 158$ (Figure 1). These small increases combine to promote the correct token as the final answer. We hypothesize that a combination of independent heuristics—termed a *bag of heuristics*—emerges across arithmetic prompts, comprising the mechanism used by the model to produce correct answers.

### 4.1 CLASSIFYING NEURONS TO HEURISTIC TYPES

To present evidence for the causal effect of the bag of heuristics on generating correct answers, we first systematically classify neurons into heuristic types. Through manual observation of key activation patterns, we identify several categories of human-identifiable heuristics, exemplified in Figure 5, and further detailed in Appendix E. To determine if a neuron $n$ at layer $l$ implements a specific heuristic, we examine the intersection between the prompts that activate the neuron and the prompts expected to be activated for this heuristic. A visual example of this procedure is shown in Figure 6. An automated algorithm of this approach is described in Appendix F.

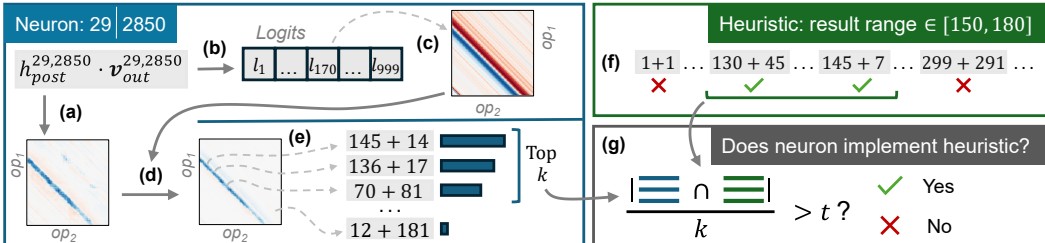

Figure 6: **Neuron to heuristic matching example.** (a) Measure the value of $h_{post}^{29,2850}$ for each operand pair $(op_1, op_2)$, using the chosen operator (addition). (b) Calculate the logits of numerical tokens embedded in $v_{out}^{29,2850}$, using Logit Lens (nostalgebraist, 2020). (c) Convert the logits vector to a 2D pattern, where the cell in index $(op_1, op_2)$ is the logit of the result token of applying the operator to $(op_1, op_2)$ (i.e. $op_1 + op_2$). (d) Multiply both patterns element-wise, to get the effective logit contribution of the neuron to the correct answer token for each prompt. (e) Extract the prompts that activate the neuron the most from the activation pattern. (f) Create a list of prompts associated with the tested heuristic. (g) Measure the intersection between the two prompt lists. If this intersection is larger than a threshold (we use $t = 0.6$), the neuron is said to implement the heuristic.

We apply this algorithm to each pair of important MLP neuron $n$ in layer $l$ and heuristic $H$. Through this method, **we classify as arithmetic heuristics 91% of the 3,200 top neurons** for each operator

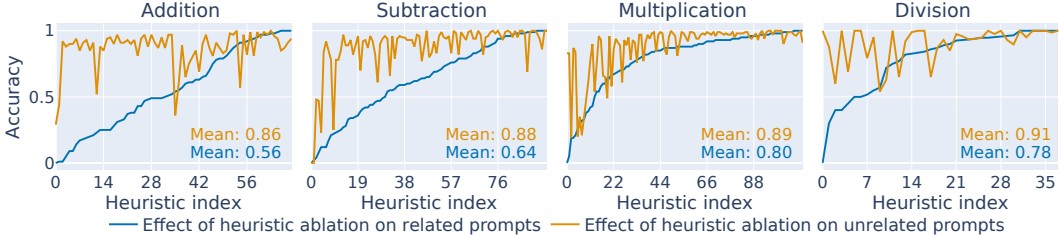

Figure 7: For each heuristic, we measure the accuracy of 100 correctly completed prompts associated with a heuristic (blue) and 100 correctly completed prompts not associated with the heuristic (yellow), after ablating that heuristic's neurons. The heuristics are sorted by the accuracy drop induced on associated prompts. Across most heuristics, ablating heuristic neurons causes a larger decrease in accuracy in prompts associated with that heuristic than in not associated prompts.

(200 per layer across 16 layers). Manual inspection of neurons that fail to classify into one of the defined heuristics reveals patterns that are not clearly identifiable (Appendix J).

## 4.2 Heuristic types are combined to answer arithmetic prompts

Following the classification of the important neurons in each middle- and late-layer MLP into one or several heuristic types, we provide evidence that the bag of heuristics is the primary mechanism the model uses to correctly answer arithmetic prompts. We show this through two ablation experiments.

**Knocking out neurons by heuristic type.** We now verify that the neurons in each heuristic type contribute to the accuracy of associated prompts by knocking out entire heuristic types and observe the resulting changes in model accuracy. We define a prompt as associated with a heuristic if and only if its components meet the conditions specified by that heuristic. (For instance, the example prompt "$226 - 68 =$" is associated with the heuristic "$op1 \equiv 0 \pmod 2$".) For each heuristic, we sample two sets of 100 correctly completed prompts each, one containing prompts associated with the heuristic and the other containing prompts not associated with that heuristic. For each heuristic, we knock out all neurons classified into it (by setting each $h_{post}^{l,n}$ activation to zero) and then remeasure the accuracy on both sets of prompts. We expect a higher decrease in accuracy on the associated prompts, since we claim each heuristic is causally linked only to its associated prompts.

The results (Figure 7) show that ablating neurons of a specific heuristic causes a significant accuracy drop on associated prompts, more than on not associated prompts, on average. This confirms the causal importance of heuristic neurons in promoting correct answer logits specifically in prompts that are associated with their heuristic type, verifying the targeted functional role of these heuristics.

However, the ablation does not result in a complete accuracy drop; it causes an average drop of 29% out of 95% average pre-ablation accuracy. We find two reasons for this. First, some heuristics have low recall: they do not apply to all associated prompts as they should (see Appendix J). Second, each prompt relies on several unrelated heuristic types, so even when one is ablated, others still contribute to increasing the correct answer's logit. In the following ablation experiment, we verify that this interplay of heuristics provides a fuller image than focusing on one heuristic at a time.

**Knocking out neurons by prompt.** To provide further evidence that the bag of heuristics is causally linked to correct arithmetic completion, we conduct a second ablation experiment. For each correct prompt, we identify the heuristic types that should affect it based on its operands and ground truth result. We then ablate the neurons with the highest classification scores (Section 4.1) in these heuristics, up to a certain neuron count, and check if the model's completion changes.

The results (Figure 8) show that ablating neurons from associated heuristics significantly drops the model's accuracy, much more than the accuracy drop caused by ablating the same number of randomly chosen neurons from unassociated heuristics. This demonstrates that we can identify the neurons important to a given prompt solely based on its associated heuristics. This also indicates a causal link between the neurons belonging to several heuristics and the prompt's correct completion. This supports our bag of heuristics claim: each heuristic only slightly boosts the correct answer logit, but combined, they cause the model to produce the correct answer with high probability.

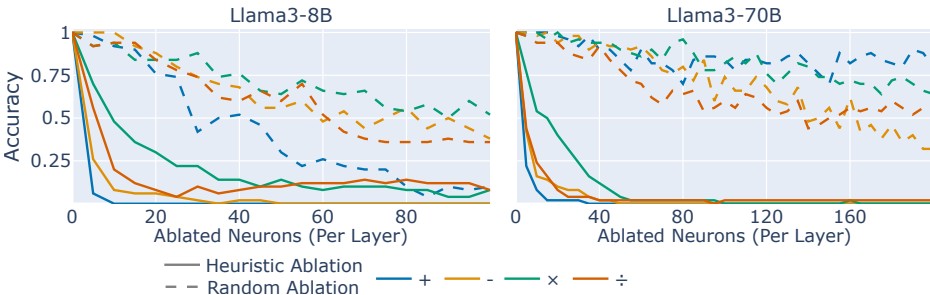

Figure 8: Knocking out neurons that implement heuristics associated with each prompt (full lines) leads to a greater decrease in accuracy than knocking out the same number of neurons whose heuristics are not associated with each prompt (dashed lines). This effect occurs across model sizes.

### 4.3 FAILURE MODES OF THE BAG OF HEURISTICS

The bag of heuristics mechanism employed in Llama3-8B does not generalize perfectly: it fails to achieve perfect accuracy across all arithmetic prompts (Appendix H). This limitation contrasts with the theoretical robustness of a genuine algorithmic approach. Here, we aim to elucidate the specific failures of this mechanism, focusing on *why* it falters for some prompts.

We hypothesize that the bag of heuristics mechanism completes prompts incorrectly in two ways. (1) The "bag" might not be big enough; i.e., a prompt might lack sufficient associated neurons. (2) the heuristics might have imperfect recall (e.g., a neuron that fires for most prompts where the first operand is even, but does not fire for the prompt "$226 - 68 =$") or have low logits for the correct answer token in the value vectors.

To test these hypotheses, we randomly sample 50 correctly completed and 50 incorrectly completed prompts. To test hypothesis (i), we count the number of heuristic neurons associated with each prompt. We find that on average, incorrect prompts have more heuristic neurons associated with them than correct prompts. Therefore, we find no support for this hypothesis. To check hypothesis (ii), we calculate the total contribution of all heuristic neurons to the logit of the correct answer for each prompt. This

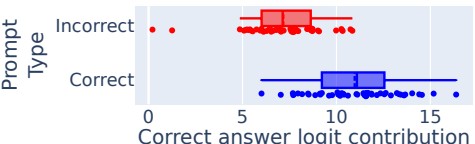

Figure 9: The model's failures can be explained by a lower total logit contribution of the heuristic neurons to the correct answers.

measurement considers both the specific activation of each neuron for the prompt, as well as the logit of the correct answer token embedded in each neuron's value vector. On average, there is indeed a slight advantage in the total logit contribution for correct prompts over incorrect prompts (Figure 9). This suggests that the primary reason for the bag of heuristics failure on certain prompts is poor promotion of correct answer logits, rather than a lack of heuristics.

## 5 TRACKING HEURISTICS DEVELOPMENT ACROSS TRAINING STEPS

Does the bag of heuristics emerge as the primary arithmetic mechanism from the onset of training, or does it override an earlier, different mechanism that initially drives arithmetic performance? We conduct an analysis of heuristic development across the training trajectory of the Pythia-6.9B model (Biderman et al., 2023), due to the public availability of its training checkpoints. Specifically, we analyze the model at its final checkpoint (143K steps) and at 10K-step intervals down to 23K steps. The 23K checkpoint is the earliest checkpoint showing good arithmetic performance; Thus, we begin our analysis at this checkpoint. To guide this analysis, we aim to answer three sub-questions:

**When do the final heuristic neurons first appear?** We examine when each heuristic neuron from the final checkpoint first appears during training. For each neuron classified into a particular heuristic type, we check if the same neuron gets classified into the same heuristic in earlier checkpoints. Averaging this measure across all heuristic types and operators provides insight into when the final heuristics initially appear during training. We observe (Figure 10a) that the model develops its final heuristic mechanism gradually across training, starting from an early checkpoint.

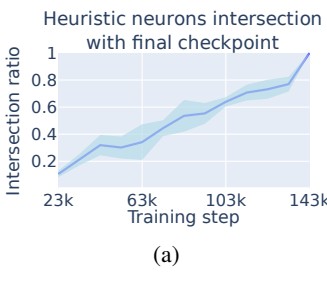 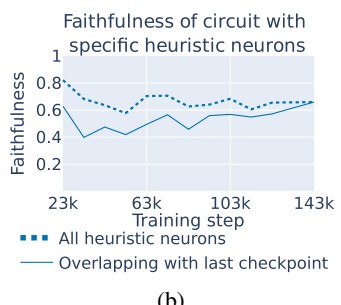 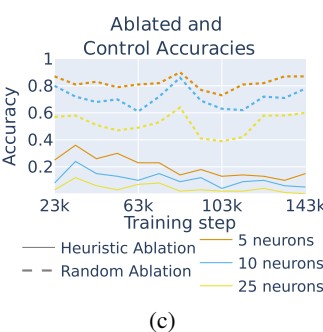

(a)          (b)          (c)

Figure 10: **Heuristic analysis across Pythia-6.9B training checkpoints.** **(a)** The percentage of heuristic neurons from the last checkpoint that also appear in previous checkpoints increases over training, revealing a gradual creation of the bag of heuristics. **(b)** The heuristic neurons that are mutual with the last checkpoint (full line) explain most of the total heuristic behavior (dashed line) at each checkpoint. Thus, the heuristics that disappear across training are less important to the model. **(c)** Ablating specific heuristic neurons heavily drops the model's accuracy across *all* training checkpoints. This suggests arithmetic accuracy primarily stems from heuristics, even in early stages.

**Do additional heuristic neurons exist mid-training?** Next, for each mid-training checkpoint, we investigate whether its heuristic neurons that are mutual with the final checkpoint make up the entire heuristic mechanism in that checkpoint, or whether other heuristics exist that later become vestigial. We examine the faithfulness (Section 2.1) of the arithmetic circuit at each checkpoint— once when including only the neurons mutual with the final checkpoint, and once when including all heuristic neurons in that checkpoint. The difference between these two measurements gives us an estimate of the importance of the mutual heuristics. Using this metric, we observe that these final heuristics explain most of the circuit performance for each intermediate checkpoint: they account for an average of 79% of the total heuristics' contribution to accuracy at each checkpoint. This indicates that, while other non-mutual heuristics exist in each checkpoint, these are less important to the circuit's accuracy and slowly become vestigial as the circuit converges to its final form.

**Does a competing arithmetic mechanism exist mid-training?** Finally, we determine if the heuristics appear as the main arithmetic mechanism from early on in training, or if they co-exist with an unrelated mechanism that becomes vestigial in later checkpoints. We repeat the prompt-guided neuron knockout experiment (Section 4.2) for each checkpoint; i.e., in each checkpoint, we sample 50 correctly completed prompts for each operator. For each prompt, we ablate 5, 10, and 25 heuristic neurons associated with the prompt in that checkpoint. We test if this targeted ablation significantly impairs the model's accuracy, even in earlier stages of training, and compare this to a baseline, where we ablate a similar amount of randomly chosen heuristic neurons. The results (Figure 10c) demonstrate that removing any amount of neurons from heuristics associated with a prompt substantially reduces the model's accuracy on these prompts even at earlier checkpoints, much more than ablating a random set of important neurons. We also observe that ablating 25 heuristic neurons per layer is enough to cause near-zero accuracy in *all* stages of training. This finding asserts that the causal link between a prompt's associated heuristics and its correct completion exists throughout training.

# 6    RELATED WORK

**Mechanistic interpretability** (MI) aims to reverse-engineer mechanisms implemented by LMs by analyzing model weights and components. Causal mediation techniques (Pearl, 2001) like activation patching (Vig et al., 2020; Geiger et al., 2021), path patching (Wang et al., 2022), and attribution patching (Nanda, 2022; Syed et al., 2023; Hanna et al., 2024b) allow localizing model behaviors to specific model components. Other studies have also presented techniques to explain the effect of specific weight matrices on input tokens (Elhage et al., 2021; Dar et al., 2023), or to analyze activations (nostalgebraist, 2020; Geva et al., 2021). Many studies have aimed to use these techniques to reverse-engineer specific behaviors of pre-trained LMs (Wang et al., 2022; Hanna et al., 2024a; Gould et al., 2024; Hou et al., 2023). We leverage MI techniques to reverse-engineer the arithmetic mechanisms implemented by pre-trained LLMs and explain them at a single-neuron resolution.

**Memorization and generalization in LLMs.** Whether models memorize training data or generalize to unseen data has been extensively studied in deep learning (e.g., Zhang et al., 2021) and specifically in LLMs (Tänzer et al., 2022; Carlini et al., 2023; Antoniades et al., 2024), but not many studies have observed this question through the lens of model internals. Among those that do, Bansal et al. (2022) attempt to predict this trade-off by observing the diversity of internal activations; Dankers & Titov (2024) show memorization in language classification tasks is not local to specific layers, and Varma et al. (2023) explain grokking using memorizing and generalizing circuits. We use this lens to observe how model internals operate in arithmetic reasoning—a task that could theoretically be solved either through extensive memorization or by learning a robust algorithm. Concurrent work (jylin et al., 2024) has shown that a LM trained to predict legal board game moves (Li et al., 2022) does so by implementing many heuristics. While heuristics would suffice to robustly predict legal moves in a board game setting, we find that the extent to which LLMs rely on heuristics is greater than prior work suggests: sets of heuristics are used to accomplish even generic tasks like arithmetic, where no heuristic is likely to generalize to all possible results.

**Arithmetic reasoning interpretability.** Recent studies on how LMs process arithmetic prompts (Stolfo et al., 2023; Zhang et al., 2024) reveal the general structure of arithmetic circuits, but do not fully explain *how* they combine operand information to produce correct answers. Our research bridges this gap by revealing the mechanism used for promoting the correct answer. Some studies show the emergence of mathematical algorithms for modular addition (Nanda et al., 2023; Zhong et al., 2024; Ding et al., 2024) and binary arithmetic (Maltoni & Ferrara, 2023) in simple, specialized toy LMs, but it is unclear if these findings extend to larger, general-purpose LMs or other operators. In pre-trained LLMs, Zhou et al. (2024) found that Fourier space features are used for addition. However, we claim this is only a partial view, as many additional types of features and heuristics relying on these features are involved in calculating answers across arithmetic operations. In this work, we give a wide view of these heuristics and how they combine to generate arithmetic answers.

## 7 CONCLUSIONS

Do LLMs rely on a robust algorithm or on memorization to solve arithmetic tasks? Our analysis suggests that the mechanism behind the arithmetic abilities of LLMs is somewhere in the middle: LLMs implement a *bag of heuristics*—a combination of many memorized rules—to perform arithmetic reasoning. To reach this conclusion, we performed a set of causal analysis experiments to locate a circuit, i.e., a subset of model components, responsible for arithmetic calculations. We examined the circuit at the level of individual neurons and pinpointed the arithmetic calculations to a sparse set of MLP neurons. We showed that each neuron acts as a memorized heuristic, activating for a specific pattern of inputs, and that the combination of many such neurons is required to correctly answer the prompts. In addition, we found that this mechanism gradually evolves over the course of training, emerging steadily rather than appearing abruptly or replacing other mechanisms.

Our results, showing LLMs' reliance on the bag of heuristics, suggest that improving LLMs' mathematical abilities may require fundamental changes to training and architectures, rather than post-hoc techniques like activation steering (Subramani et al., 2022; Turner et al., 2023). Additionally, the evolution of this mechanism across training indicates that models learn these heuristics early and reinforce them over time, potentially overfitting to early simple strategies; it is unclear if regularization can improve this, and this is a possible avenue for future research.

## 8 LIMITATIONS AND DISCUSSION

Interpretability work is often fundamentally limited by human biases. As researchers, we often impose human abstractions onto models, whereas the goal of interpretability is to understand the abstractions that models learn and apply in a way that we can understand. Our work is also subject to this limitation, namely with respect to the definition of heuristic types: We define heuristic abstractions based on our human-identifiable definitions. A possible improvement would be to develop a method to identify these abstractions without human bias. Another important detail is that our analysis focuses on LLMs that combine digits in tokenization. That is, every token can contain more than one digit. The robust algorithms used by humans depend on our ability to separate larger numbers to single digits. Thus, a similar analysis might lead to different conclusions for models that perform single-digit tokenization.

ACKNOWLEDGMENTS

We are grateful to Dana Arad and Alessandro Stolfo for providing feedback for this work. This research was supported by the Israel Science Foundation (grant No. 448/20), an Azrieli Foundation Early Career Faculty Fellowship, and an AI Alignment grant from Open Philanthropy. AM is supported by a postdoctoral fellowship under the Zuckerman STEM Leadership Program. AR is supported by a postdoctoral fellowship under the Azrieli Postdoctoral Fellowship Program. This research was funded by the European Union (ERC, Control-LM, 101165402). Views and opinions expressed are however those of the author(s) only and do not necessarily reflect those of the European Union or the European Research Council Executive Agency. Neither the European Union nor the granting authority can be held responsible for them.

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

## A    LLAMA3-8B CIRCUIT EVALUATION RESULTS

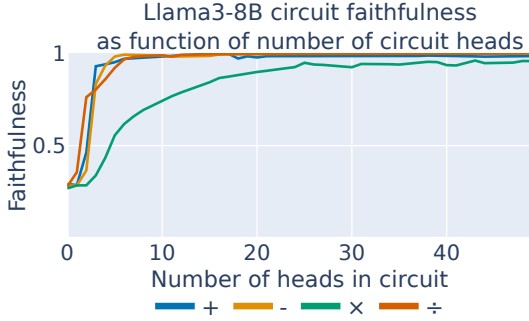

Figure 11: Llama3-8B arithmetic circuit faithfulness as function of number of circuit heads.

In Figure 11, we present an analysis of the faithfulness of the identified circuit in Llama3-8B (Section 2.1) as a function of the number of attention heads within the circuit. The circuit includes all MLPs (the neuron-level analysis in Section 3.1 does not apply in this context). Our goal is to find a minimal subset of the model with the fewest attention heads possible while maintaining high faithfulness. We assess faithfulness independently for each operator to obtain a more nuanced understanding of the necessary attention heads. For addition, subtraction, and division operations, we observe that 6 heads suffice to attain high faithfulness (97% on average). In contrast, in multiplication, 20 attention heads are required to achieve a faithfulness score exceeding 90%. The faithfulness of the full arithmetic circuit for each operator, corresponding to the Pareto-optimal number of heads (in terms of achieving high faithfulness with as few heads as possible), is documented in Table 1. We explore the attention patterns of these attention heads in the next section.

Table 1: Llama3-8B arithmetic circuit faithfulness, per operator.

| Operator | + | - | × | ÷ |
|---|---|---|---|---|
| **Faithfulness** | 0.97 | 0.98 | 0.90 | 0.96 |
| **# Attn Heads** | 6 | 6 | 20 | 6 |

## B    LLAMA3-8B CIRCUIT ADDITIONAL COMPONENTS

To provide a more comprehensive understanding of the arithmetic circuit in Llama3-8B, we analyze the additional components that compose it, namely the first MLP layer (MLP0) and the high-effect attention heads.

### B.1    MLP0

To analyze the role of MLP0 in the arithmetic circuit, we first test if—similarly to the middle- and late-layer MLPs—only a sparse set of neurons within the MLP is required. We measure the intervention effect of each neuron, averaged across the operands and operator positions ($p \in [1, 2, 3]$). The results, shown in Figure 12a, reveal that few MLP0 neurons have a high effect, similar to the middle- and late-layer MLPs. To verify these neurons are sufficient for arithmetic calculations, we repeat the experiment from Section 3.1. Specifically, we measure the faithfulness of the circuit—consisting of the top 1% of neurons in each of the middle and late layers ($l \in [16, 32]$) as well as a varying number of neurons in MLP0. The results, shown in Figure 12b, reveal that also in the first MLP layer, as little as 1% of neurons is sufficient for the circuit to achieve high faithfulness, similarly to the middle- and late-layer MLPs.

Because Llama3-8B applies a positional embedding only at attention layers, the activation of MLP0 is not affected by the position of any token. Additionally, due to a lack of attention heads that move information between the operand and operator positions before MLP0, we can analyze its effect directly on single tokens. Thus, we view MLP0 as an "effective embedding" (McDougall et al.,

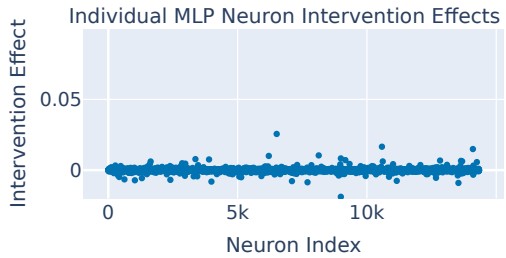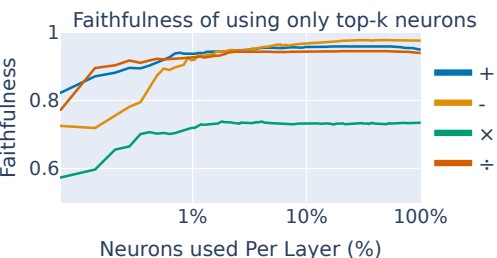

(a) Few neurons in the first MLP have a high effect on arithmetic prompts.

(b) The circuit faithfulness when including a percentage of high-effect MLP0 neurons in the circuit.

Figure 12: Analyzing effect of individual MLP0 neurons. A small number of neurons is sufficient for circuit accuracy.

2023), and hypothesize that the role of each MLP0 neuron is to incorporate additional numerical information into each token embedding. To verify this, we pass a list of numerical tokens, each representing an operand in our analyzed operand range ($t \in [0, 300]$), through the model. We measure the activation of each high-effect neuron for each token. As exemplified in Figure 13, the activations of these high-effect neurons correspond to varied numerical features. For example, Neuron 6206 (Figure 13a activates for numbers near 170 or 17; Neuron 7101 activates for numbers greater than 100; Neuron 8969 activates for numbers that are congruent to $8 \pmod{10}$. Overall, the set of patterns identified by these neurons can be used by the middle- and late-layer heuristic neurons to perform their more complex functionalities.

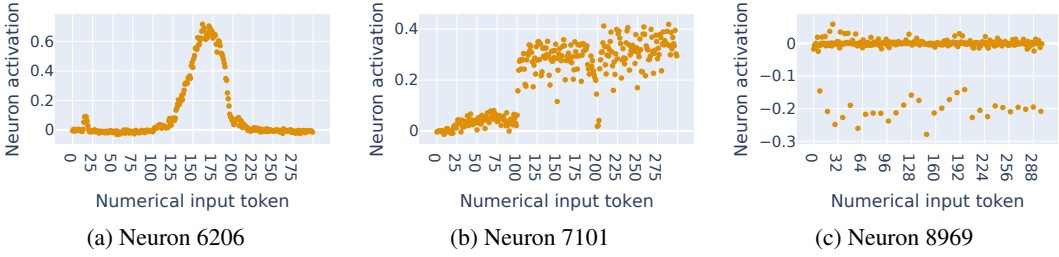

(a) Neuron 6206          (b) Neuron 7101          (c) Neuron 8969

Figure 13: Individual MLP0 neurons have identifiable activation patterns for numerical tokens.

## B.2 ATTENTION HEAD PATTERNS

We present the attention patterns of the attention heads that are contained in the arithmetic circuit. In Section 2.1, for simplicity, we consider the arithmetic circuit as a single circuit for all arithmetic operators, containing all attention heads used across the four operators. We observe that to achieve high faithfulness of the circuit with a minimal number of heads (Appendix A), some general arithmetic heads, that significantly improve faithfulness across all operators, are required. Other heads, while contributing to increased faithfulness, are operator-specific. They exhibit varying levels of importance for different operators. A full description of the attention heads used in the arithmetic circuit for each operator is provided in Table 2.

To better understand the role of the general arithmetic heads (L2H2, L15H13, L16H21), we compute their attention patterns, averaging them across our prompt dataset (Section 2.1). These patterns (Figure 14) reveal a clear signal: each of the three heads attends to a single input token, copying the representation from that position and projecting it to the last position. Specifically, L16H21 attends to the first operand, L2H2 attends to the operator and L15H13 attends to the second operand. This implies the role of each such head is to move the representation from that position, which includes to the last position, where it is further processed by the bag of heuristics implemented in the middle- and late-layer MLPs. To confirm that the information copied from each position to the last position consists solely of data from the token at that position, we conduct an ablation study. For each general arithmetic head, we zero out all preceding attention patterns that move information

Table 2: Llama3-8B operator-specific arithmetic circuit attention heads. The general arithmetic heads are marked **bold**. L$i$H$j$ denotes the $j^{\text{th}}$ attention head in Layer $i$.

| Operator | Circuit Heads |
|---|---|
| $+$ | **L2H2**, L5H3, L5H31, L14H12, **L15H13**, **L16H21** |
| $-$ | **L2H2**, L13H21, L13H22, L14H12, **L15H13**, **L16H21** |
| $\times$ | **L2H2**, L5H30, L8H15, L9H26, L13H18, L13H21, L13H22, L14H12, L14H13, L15H8, **L15H13**, L15H14, L15H15, L16H3, **L16H21**, L17H24, L17H26, L18H16, L20H2, L22H1 |
| $\div$ | **L2H2**, L5H31, **L15H13**, L15H14, **L16H21**, L18H16 |

*to* its attended position (e.g., for L15H13 we zero out all patterns that move information to the $op_2$ position), preventing any influence from previous positions. We observe this does not affect the circuit's performance, indicating that the representation copied by each general arithmetic head to the final position contains information only regarding the token at its original position.

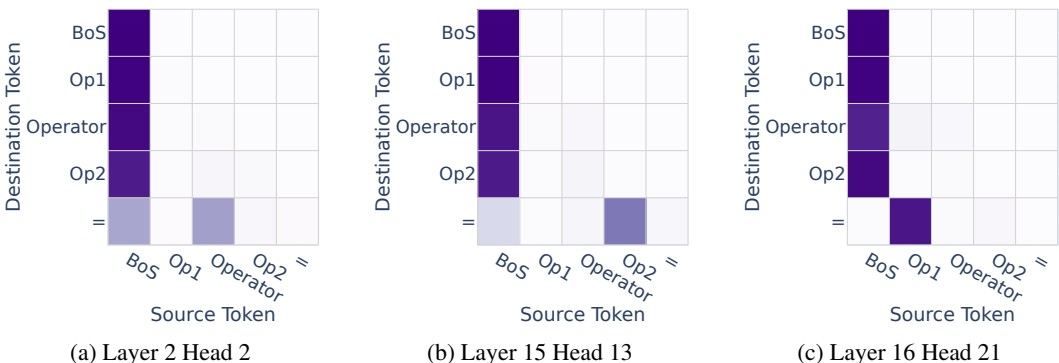

(a) Layer 2 Head 2     (b) Layer 15 Head 13     (c) Layer 16 Head 21

Figure 14: The attention patterns for the general arithmetic attention heads in Llama3-8B, show that each head attends, at the last token, to a single previous token across all prompts. When combined, these heads attend to all three operand and operator positions.

## C  MULTI-LAYER PERCEPTRON IMPLEMENTATION DETAILS

Across the models we analyze, different implementations exist for the MLP layer in a transformer block. More specifically, Pythia-6.9B and GPT-J use simple MLP layers, consisting of two matrix multiplications. For a layer $l$, it can be described as:

$$\mathbf{h}_{post}^l = \sigma\left(\mathbf{h}_{in}^l \mathbf{W}_{in}^{l\top}\right) \tag{4}$$

$$\mathbf{h}_{out}^l = \mathbf{h}_{post}^l \mathbf{W}_{out}^l \tag{5}$$

where $\mathbf{h}_{in}^l, \mathbf{h}_{out}^l \in \mathbb{R}^d$ are the input and output representations of the MLP at layer $l$, respectively. $\mathbf{h}_{post}^l \in \mathbb{R}^{d_{mlp}}$ is the post-activation vector, $\mathbf{W}_{in}^l, \mathbf{W}_{out}^l \in \mathbb{R}^{d_{mlp}\times d}$ are parameter matrices, and $\sigma$ is a non-linearity function. Llama3-8B and Llama3-70B use a Gated MLP layer (Liu et al., 2021), described in the following two equations for layer $l$:

$$\mathbf{h}_{post}^l = \sigma(\mathbf{h}_{in}^l \mathbf{W}_{gate}^{l\top}) \circ (\mathbf{h}_{in}^l \mathbf{W}_{in}^{l\top}) \tag{6}$$

$$\mathbf{h}_{out}^l = \mathbf{h}_{post}^l \mathbf{W}_{out}^l \tag{7}$$

where $\mathbf{W}_{gate}^l \in \mathbb{R}^{d_{mlp}\times d}$ is an additional parameter matrix and $\circ$ is Hadamard product. Biases are omitted in both presentations.

16

While the key-value view that was used in Section 3 was devised for simple MLPs (Geva et al., 2021), it can be applied to the Gated MLP mechanism of Llama3 models as well. For the $n^{\text{th}}$ neuron, we treat the element $\mathbf{h}_{post}^{l,n}$ of $\mathbf{h}_{post}^{l}$ as the activation of the $n^{\text{th}}$ key vector. Each such activation is multiplied with $\mathbf{v}_{out}^{l,n}$, a row vector of $\mathbf{W}_{out}^{l}$, resulting in the same multiplication as performed in simple MLPs.

## D NEURON INTERSECTION BETWEEN OPERATORS

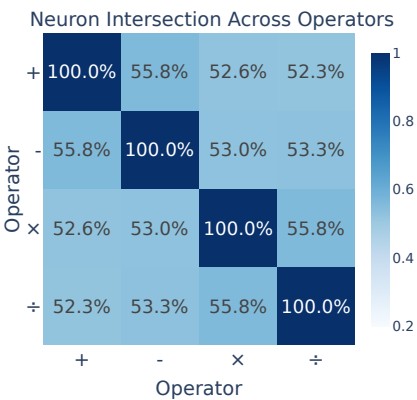

Figure 15: IoU of MLP neurons between operator-specific circuits

Figure 15 presents the IoU (intersection over union) of causally important neuron sets in the arithmetic circuits, identified separately for each operator. Each circuit consists of the top-200 neurons from each layer $l \in [16, 32]$, ranked based on their mean effect (Section 3.1), totaling $3,200$ neurons. The average IoU between any two operator-specific circuits is relatively low (54%), indicating that a substantial proportion of key neurons are unique to each operator. Consequently, we define a distinct circuit for each arithmetic operator at the neuron level (Section 3.1).

## E HEURISTIC TYPES DESCRIPTIONS

In this section, we present the various heuristic types that were manually identified.

Most heuristic types are defined with parameters. A neuron is defined as a heuristic $H$, if it matches the heuristic definition given a parameter value. For example, the neuron in Figure 1b is a range heuristic with a range parameter $[150, 180]$. In Algorithm 1, we describe the process of matching between each important neuron and each heuristic $H$. The parameters are sub-sampled evenly to cover most observed possibilities, which causes some inaccuracies (for example, a neuron that fires when $op_2 \in [105, 205]$ will be classified as a range heuristic with $op_2 \in [100, 200]$ due to our parameter choices).

Several types of heuristics (Range, Modulo, Pattern) can apply either to an operand in a prompt or to the prompt's result. For example, an "operand range" heuristic is triggered when $op_1$ or $op_2$ falls within a specific numerical range, while a "result range" heuristic activates when the prompt's ground truth result is within a defined range.

The full list of heuristic types is as follows:

- **Range heuristic:** A neuron that activates when a value (either an operand or result) falls within a specified range $[a, b]$. The parameters $a, b \in \mathbb{N}$ are chosen such that the range length $len = (b - a) \in [10, 30, 50, 100]$ for addition, subtraction, and multiplication, and $len = (b - a) \in [2, 10, 100]$ for division (due to a different distribution of potential results for integer division). The range start, $a$, is defined as $a \in x_n : x_n = x_0 + n \cdot \max(\lfloor \frac{len}{3} \rfloor, 10), n \in \mathbb{N}, x_0 = 0$, provided $a$ remains below the maximum prompt result. This definition of range start and length allows for ranges to intersect or overlap.

- **Modulo heuristic:** A neuron that activates when a value (either operand or result) is congruent to $m$ modulo $n$. The parameters are $n \in \{2, 3, 4, 5, 6, 7, 8, 9, 11, 13, 15\}$ and $m \in [0, n-1]$. We exclude $n = 10$ or $n = 100$ as these are specific cases of the broader "pattern" heuristics, discussed next. Additionally, $n = 12, 14$, and $n \geq 16$ are excluded, as no neurons were classified under these parameter values.

- **Pattern heuristic:** A neuron that activates when a value (either operand or result) matches a specific regular expression $p$. The parameter $p$ is a 3-digit regular expression, potentially padded with leading zeros. For example, a neuron classified under the "operand pattern 1.2" heuristic activates when one of the operands has 1 in the hundreds place and 2 in the units place.

- **Identical operands heuristic:** A neuron that activates when both operands are equal ($op_1$ == $op_2$). The tokens embedded in the value vector change depending on the operator for which the neuron activates. For instance, in subtraction, such neurons have been observed to promote the "0" token (the result of subtracting a number from itself).

- **Multi-result heuristic:** A neuron that promotes a set $\mathbf{S}$ of several unrelated results (For example, a neuron that promote the results $[4, 5, 7]$). This heuristic type is defined exclusively for division, where the parameter $S$ can consist of several values, where $|\mathbf{S}| \in [2, 4]$. This range was chosen based on observations of activation patterns. The values included in $\mathbf{S}$ are chosen according to the result tokens that the neuron contributes the most to: we determine $\mathbf{S}$ by examining the top prompts for which the neuron promotes their answer. From these top answers, we identify a minimal set of 2 to 4 distinct result values. This set of result values is chosen if it accounts for more than a threshold percentage of the results. This approach captures the most significant result values that the neuron consistently promotes in division operations. If no such set of different results exists, a neuron is not classified as this heuristic.

## F   HEURISTIC NEURON CLASSIFICATION ALGORITHM

We present the full algorithm used to match between each neuron and each heuristic type in Algorithm 1 (as exemplified in Figure 6). The goal of this process is to check if a neuron implements a specific heuristic. We repeat this process for each pair of neuron $n$ in layer $l$ and heuristic $H$. This method allows a single neuron to be classified as several heuristics. The matching of each neuron to heuristics is done separately for each arithmetic operator.

---

**Algorithm 1** Neuron Classification To Heuristic Type

**Inputs:** Heuristic $H$, Layer $l \in [1, l_{max}]$, Neuron Index $n \in [1, n_{max}]$,
         Operator $o \in \{+, -, \times, \div\}$, Threshold $t \in [0, 1.0]$

1: $\mathbf{A} \leftarrow \text{GenerateActivationPattern}(l, n, o)$  ▷ 2D Activation pattern
2: **if** $H$ is a direct heuristic **then**
3:     $\mathbf{l} \leftarrow \text{LogitLens}(\mathbf{v}_{out}^{l,n})$  ▷ Logits over vocabulary
4:     $\mathbf{L} \leftarrow \text{ConvertToPattern}(\mathbf{l})$  ▷ 2D Logits Pattern
5:     $\mathbf{A} \leftarrow \mathbf{A} \circ \mathbf{L}$  ▷ Element-wise multiplication
6: **end if**
7: $prompts_{heuristic} \leftarrow \text{GetAssociatedPrompts}(H)$  ▷ Expected prompts to activate in $H$
8: $k \leftarrow |prompts_{heuristic}|$
9: $prompts_{neuron} \leftarrow \text{GetTopKActivatingPrompts}(\mathbf{A}, k)$  ▷ Prompts that activate neuron
10: $prompts_{intersection} \leftarrow prompts_{heuristic} \cap prompts_{neuron}$
11: $score \leftarrow \dfrac{|prompts_{intersection}|}{k}$  ▷ Normalize intersection to $[0, 1.0]$
12: **return** $score \geq t$

---

The algorithm first calculates the neuron's activations $h_{post}^{l,n}$ for all prompts, yielding a 2D activation pattern (Line 1), as those seen in Appendix J. When checking if a neuron implements a *direct* heuristic (heuristics that directly promote relevant result tokens (Section 3.2)), we also need to take into consideration the tokens that are embedded in the neuron's value vector $\mathbf{v}_{out}^{l,n}$. This is not done in indirect heuristics because we do not expect relevant result tokens to be promoted in such heuristics. Thus, we extract the logits of all numerical tokens ("0", "1", "2", ... "999") from the value vector

$\mathbf{v}_{out}^{l,n}$ (Line 3), using Logit Lens (nostalgebraist, 2020). To match the logit of each result token to the prompts that result in it (e.g., match the logit of "151" with the prompts "$1 + 150 =$","$2 + 149 =$", etc.), we convert the logits to a 2D pattern $\mathbf{L}$, where $\mathbf{L}_{ij}$ is the logit of applying the operator on $i$ and $j$ (Line 4). Multiplying the 2D activation pattern and 2D logit pattern element-wise (Line 5) yields the neuron's effective logit contribution to the correct answer of each prompt, i.e., the value at each index $i, j$ marks how much the neuron promotes the answer token to the result of applying the operator on $i$ and $j$. We then create two lists of prompts. One list contains prompts associated with the heuristic (Line 7), to be used as the "ground truth". We mark the number of associated prompts as $k$ (Line 8), and find a second list, containing the top-$k$ prompts that are most contributed to by the neuron (Line 9). We measure if the intersection of these two lists, normalized by $k$, is larger than the threshold $t = 0.6$ (Line 10–Line 12) to decide if the neuron implements the heuristic.

## G  ADDITIONAL IMPLEMENTATION DETAILS

All experimental procedures were executed using the TransformerLens library Nanda & Bloom (2022). Experiments involving Llama3-8B, GPT-J, and Pythia-6.9B were conducted on a single Nvidia-L40 GPU with 48GB of GPU memory. Llama3-70B was loaded in 16-bit precision across four Nvidia A100 GPUs, each with 80GB of GPU memory.

**Arithmetic prompts.**  Unless otherwise specified, our experiments utilize prompts of the form $op_1 \circ op_2 =$", where $op_1, op_2 \in [0, 300]$ and $\circ \in \{+, -, \times, \div\}$. The division operator ($\div$) is interpreted as integer division, considering only the integer part of the result (e.g., the ground truth result for $45 \div 4 =$" is 11). For each prompt, we compute the ground-truth result and exclude prompts whose results are not tokenized to a single token. This filtering process eliminates prompts with negative results and prompts for which the correct answer exceeds the single-token limit (The highest token from which the model splits numbers into two tokens). The single-token limit $s$ for Llama3-8B and Llama3-70B is $s_{llama3} = 1,000$, while for GPT-J and Pythia-6.9B, it is $s_{gptj} = s_{pythia} = 520$.

**Circuit discovery via activation patching.**  In Section 2.1, we quantify the effect of patching on probabilities. We find that applying our effect measure (Equation (1)) on logits instead of probabilities, does not alter the result significantly—the most effective components remain consistent, differing only in the scale of the effect.

**Circuit faithfulness evaluation.**  Thus far, to measure the faithfulness of the circuit, we have performed mean ablation on all non-circuit components. For that, we calculate the mean activation output of each component, across all arithmetic prompts. In this context, we do not filter correct prompts exclusively, but instead use *all* prompts of the form "$op_1 \circ op_2 =$", for $op_1, op_2 \in [0, 300]$. This leads to an equal amount of prompts per operator, thus maintaining the mean activation balanced across the different operators. When evaluating a circuit with partial MLP layers (Section 3.1), we mean ablate lower-effect neurons in middle- and late-layer MLPs only, starting and ending in the earliest and latest layers from which the answer is extractable via linear probe (Section 2.2). These layer ranges are $[16, 32], [39, 80], [14, 32], [17, 28]$ In Llama3-8B, Llama3-70B, Pythia-6.9B, and GPT-J, respectively.

**Linear probing for correct answers.**  We define a linear probe $f_{l,p} : \mathbb{R} \to \mathbb{R}^s$, where $s$ is the single-token limit, to predict the correct answer token from the output representation at layer $l$ and position $p$. We compute these outputs by using all correctly completed prompts, with 80% used for training the classifier and 20% for evaluation. Each probing classifier is implemented as a one-layer fully connected model. We train it using the Adam optimizer (Kingma & Ba, 2015) with a learning rate of 0.0003 and a batch size of 32, optimizing a cross-entropy loss function.

**Heuristic classification.**  In applying the heuristic classification algorithm (Section 4.1), we use all activations from the last position $p = 4$ and employ a classification score threshold $t = 0.6$. Due to a lack of ground truth data for neuron-to-heuristic matches, the threshold is chosen to achieve a Pareto-optimal balance of false positives, that occur more for a lower threshold, and false negatives, that occur more in a higher threshold. The amount of false classifications is estimated manually.

Table 3: Accuracy of the analyzed models on arithmetic prompts.

| Model | Operator | | | | Average |
|---|---|---|---|---|---|
| | $+$ | $-$ | $\times$ | $\div$ | |
| Llama3-8B | 0.97 | 0.96 | 0.84 | 0.92 | 0.95 |
| Llama3-70B | 0.97 | 0.99 | 0.99 | 0.73 | 0.88 |
| Pythia-6.9B | 0.30 | 0.04 | 0.27 | 0.75 | 0.43 |
| GPT-J | 0.23 | 0.09 | 0.46 | 0.64 | 0.37 |

## H  MODEL ACCURACIES ON ARITHMETIC PROMPTS

Table 3 presents the accuracy of the analyzed models on arithmetic prompts. The accuracy is evaluated across prompts where the result is represented by a single token, with both operands constrained to the interval $[0, 300]$, where 300 represents the maximum value of operands in the analyzed prompts (Section 2.1), chosen for efficiency. It is noteworthy that the high accuracy rates observed in division operations for the smaller-scale models (GPT-J, Pythia-6.9B) can be attributed to the non-uniform distribution of answers in integer division, i.e. - half of the legal prompts result in the token '0'.

## I  RESULTS ON ADDITIONAL MODELS

To demonstrate the generalizability of our findings across differently-trained LLMs, we conduct our primary experiments on Llama3-70B (Dubey et al., 2024), Pythia-6.9B (Biderman et al., 2023), and GPT-J (Wang & Komatsuzaki, 2021). We replicate the experiments for circuit discovery and evaluation (Section 2.1), linear probing for answer token embeddings (Section 2.2), top-k neuron faithfulness analysis (Section 3.1), and heuristic analysis (Section 4.2). The results obtained are similar to those of Llama3-8B across all three LLMs:

- The circuit comprises a sparse subset of attention heads that project operand and operator information to the final position, along with all MLP layers. Early-layer MLPs process information at the operand and operator positions, while middle- and late-layer MLPs process the combined information at the last token. These findings are illustrated for Llama3-70B (Figure 16a), Pythia-6.9B (Figure 17a), and GPT-J (Figure 18a).

- Linear probing results indicate that the correct answer can only be extracted with high accuracy in the final position, following the processing initiated by middle-layer MLPs. These observations are presented for Llama3-70B (Figure 16b), Pythia-6.9B (Figure 17b), and GPT-J (Figure 18b).

- The circuit requires a sparse subset of middle- and late-layer MLP neurons to achieve maximal faithfulness. These results are depicted for Llama3-70B (Figure 16c), Pythia-6.9B (Figure 17c), and GPT-J (Figure 18c).

- The heuristic neurons in the middle- and late-layer MLPs are the model components that write the correct answer to the model output. This is shown by repeating the prompt-guided knockout experiment (Section 4.2). The ablation of specific heuristic neurons associated with prompts results in a more significant reduction in model accuracy compared to the ablation of a random set of neurons of equivalent size. These findings are illustrated for Llama3-70B (Figure 8), Pythia-6.9B (Figure 17d), and GPT-J (Figure 18d).

The results observed in Llama3-70B exhibit the highest similarity to those reported in Llama3-8B, with a more pronounced knockout effect compared to the other two models. We hypothesize that this indicates a more sophisticated development of the bag of heuristics in Llama3-70B, potentially facilitated by its larger size and modern training methodology.

## J  ADDITIONAL EXAMPLES FOR ARITHMETIC HEURISTICS

We present supplementary examples of heuristic neurons (Table 4, Table 5, Table 6, and Table 7 for the four arithmetic operators, respectively). Each neuron $\mathbf{h}_{post}^{l,n}$ found to be important for a specific

arithmetic operator (Section 3.1) is categorized as one or more heuristics. For each designated operator, we randomly present four neurons. We compute and report each neuron's activation pattern, the ten numerical tokens with the strongest embeddings in each neuron's value vector $\mathbf{v}_{out}^{l,n}$, and provide a non-exhaustive list of the heuristics it implements.

Furthermore, we present in Table 8 several examples of causally significant neurons that have not been classified as specific heuristics. These neurons are relevant to the aforementioned limitation in our methodology (Section 8); it is conceivable that they too may be considered components of the bag of heuristics, depending upon our ability to comprehend the abstractions they implement.

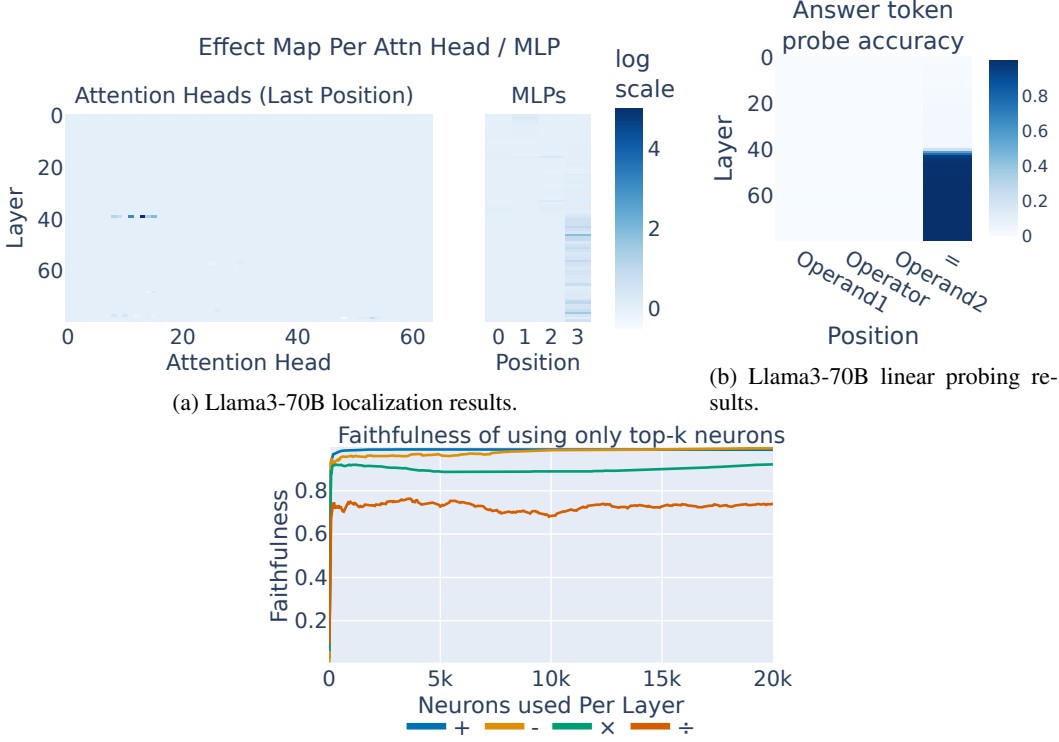

(a) Llama3-70B localization results.

(b) Llama3-70B linear probing results.

(c) Llama3-70B faithfulness as function of MLP neurons in each layer.

Figure 16: Llama3-70B analysis results

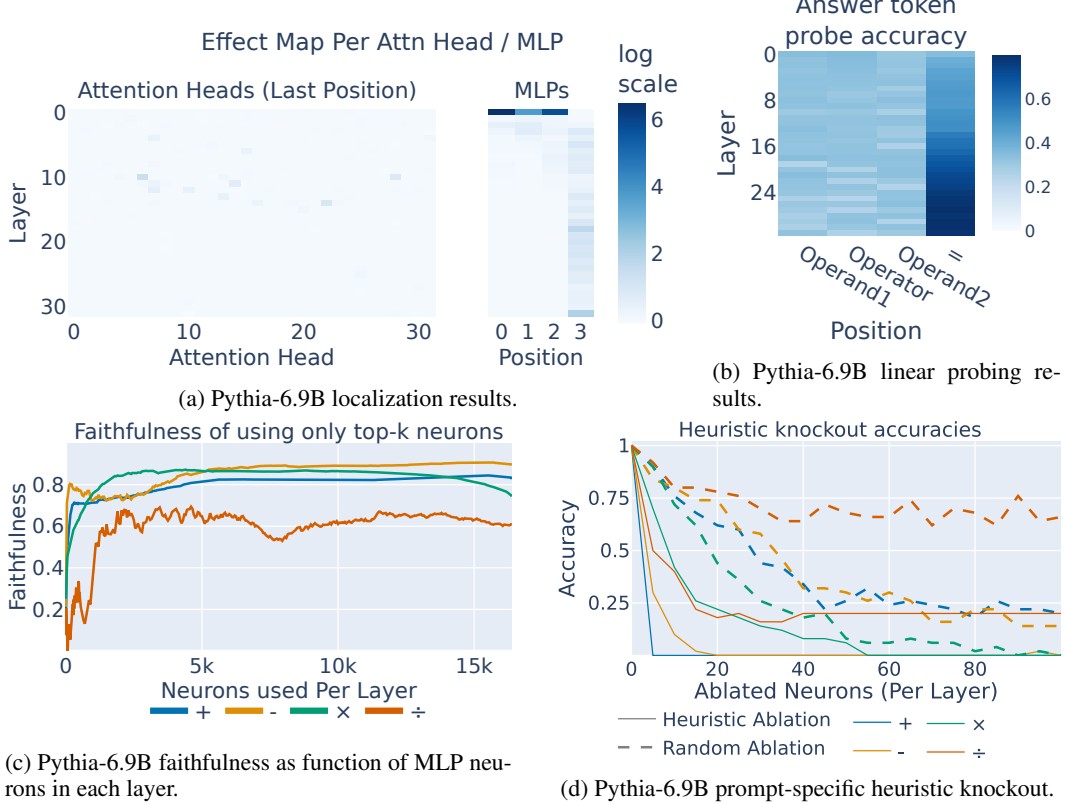

(a) Pythia-6.9B localization results.

(b) Pythia-6.9B linear probing results.

(c) Pythia-6.9B faithfulness as function of MLP neurons in each layer.

(d) Pythia-6.9B prompt-specific heuristic knockout.

Figure 17: Results for all analyses on Pythia-6.9B.

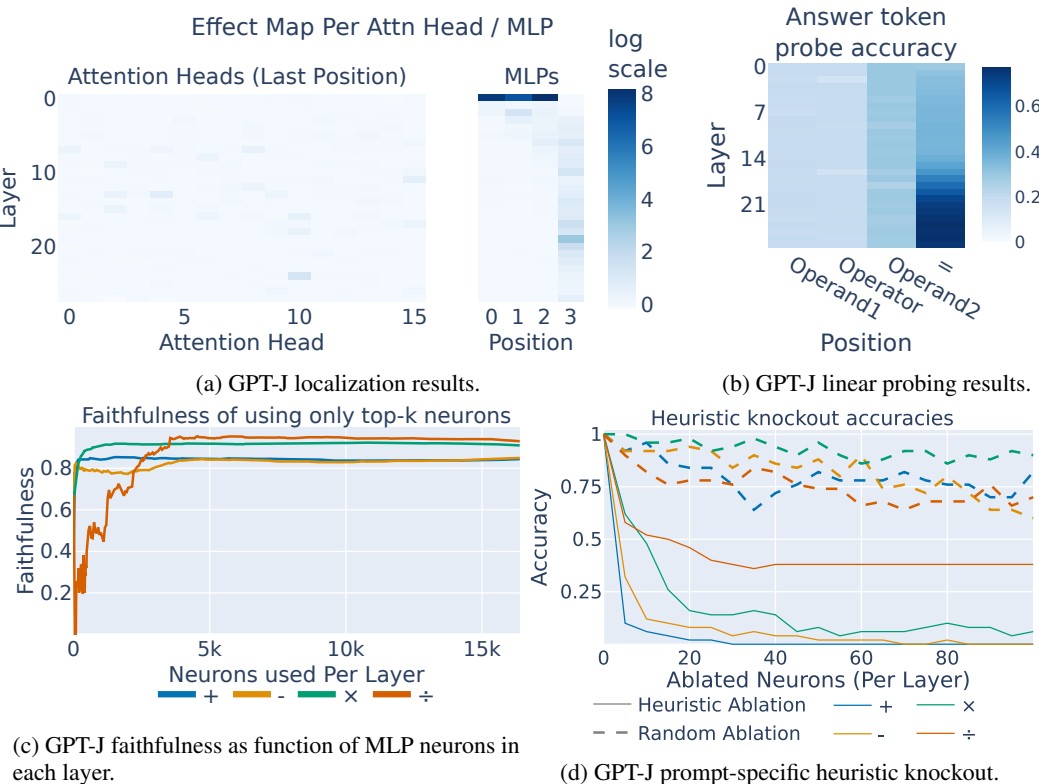

(a) GPT-J localization results.

(b) GPT-J linear probing results.

(c) GPT-J faithfulness as function of MLP neurons in each layer.

(d) GPT-J prompt-specific heuristic knockout.

Figure 18: Results for all analyses on GPT-J.

Table 4: Examples of heuristic neurons for the addition operator

| Neuron number $\mathbf{h}_{post}^{l,n}$ | Activation pattern | Top-10 numerical logits in value vector $\mathbf{v}_{out}^{l,n}$ | Classified Heuristics |
|---|---|---|---|
| $\mathbf{h}_{post}^{31,9208}$ |  | '254', '255', '253', '256', '250', '257', '252', '251', '258', '259' | $\text{range}_{result} \in [240, 270]$, $\text{range}_{result} \in [250, 300]$, $\text{pattern}_{result} .5.$ |
| $\mathbf{h}_{post}^{31,2919}$ |  | '164', '163', '464', '363', '63', '063', '364', '963', '263', '663' | $\text{pattern}_{result} .6.$, $\text{pattern}_{result} .64$ |
| $\mathbf{h}_{post}^{17,1463}$ |  | '2', '542', '792', '979', '372', '502', '882', '962', '602', '02' | $op1 \equiv 2 \pmod 5$, $op2 \equiv 2 \pmod 5$ |
| $\mathbf{h}_{post}^{16,10226}$ |  | '922', '169', '322', '258', '873', '753', '129', '852', '41', '33' | $\text{range}_{op1} \in [0, 50]$, $\text{range}_{op1} \in [0, 100]$, $\text{range}_{op2} \in [0, 100]$ |

Table 5: Examples of heuristic neurons for the subtraction operator

| Neuron number $\mathbf{h}_{post}^{l,n}$ | Activation pattern | Top-10 numerical logits in value vector $\mathbf{v}_{out}^{l,n}$ | Classified Heuristics |
|---|---|---|---|
| $\mathbf{h}_{post}^{30,8806}$ |  | '0', '000', '00', '006', '002', '004', '001', '005', '003', '009' | identical operands, $\text{pattern}_{result}$ 000 |
| $\mathbf{h}_{post}^{29,1647}$ |  | '192', '384', '288', '576', '336', '672', '96', '224', '608', '768' | $result \equiv 0 \pmod 8$ |
| $\mathbf{h}_{post}^{16,747}$ |  | '15', '5', '015', '535', '764', '3', '762', '021', '4', '13' | $\text{pattern}_{result}$ 0.5 |
| $\mathbf{h}_{post}^{16,360}$ |  | '369', '271', '761', '762', '522', '918', '972', '783', '325', '531' | $\text{pattern}_{op2}$ ..8 |

Table 6: Examples of heuristic neurons for the multiplication operator

| Neuron number $\mathbf{h}_{post}^{l,n}$ | Activation pattern | Top-10 numerical logits in value vector $\mathbf{v}_{out}^{l,n}$ | Classified Heuristics |
|---|---|---|---|
| $\mathbf{h}_{post}^{31,8084}$ |  | '320', '360', '770', '630', '820', '510', '420', '240', '370', '380' | $result \equiv 0 \pmod 5$, $\text{pattern}_{result} ..5$, $\text{pattern}_{op2} 010$ |
| $\mathbf{h}_{post}^{30,9988}$ |  | '585', '936', '546', '637', '975', '455', '910', '819', '715', '364' | $op1 \equiv 0 \pmod{13}$, $op2 \equiv 0 \pmod{13}$, $result \equiv 0 \pmod{13}$ |
| $\mathbf{h}_{post}^{21,12521}$ |  | '170', '162', '171', '169', '17', '156', '160', '16', '785', '168' | $\text{range}_{result} \in [150, 180]$, $result \equiv 0 \pmod 2$ |
| $\mathbf{h}_{post}^{16,2956}$ |  | '844', '848', '873', '985', '548', '858', '657', '995', '788', '716' | $\text{pattern}_{op2} 0.1$ |

Table 7: Examples of heuristic neurons for the division operator

| Neuron number $\mathbf{h}_{post}^{l,n}$ | Activation pattern | Top-10 numerical logits in value vector $\mathbf{v}_{out}^{l,n}$ | Classified Heuristics |
|---|---|---|---|
| $\mathbf{h}_{post}^{30,3168}$ |  | '5', '005', '05', '948', '891', '349', '324', '544', '905', '801' | $\text{pattern}_{result}\,005$, $\text{range}_{result} \in [4-6]$ |
| $\mathbf{h}_{post}^{25,1402}$ |  | '23', '33', '38', '08', '22', '00', '26', '30', '07', '32' | $\text{range}_{op2} \in [0,10]$, $\text{range}_{result} \in [0,100]$, $\text{pattern}_{result}\,..3$ |
| $\mathbf{h}_{post}^{24,9306}$ |  | '8', '446', '770', '7', '740', '674', '449', '741', '08', '433' | $\text{multi-value}_{result} \in \{1,2,7,8\}$ |
| $\mathbf{h}_{post}^{17,10704}$ |  | '568', '316', '773', '405', '621', '822', '582', '549', '587', '598', | $\text{pattern}_{op1}\,..0$, $\text{pattern}_{op1}\,.50$ |

Table 8: Examples of neurons that failed to classify as one of the defined heuristic types

| Neuron number $\mathbf{h}_{post}^{l,n}$ | Activation pattern | Top-10 numerical logits in value vector $\mathbf{v}_{out}^{l,n}$ |
|---|---|---|
| $\mathbf{h}_{post}^{22,14024}$, Addition |  | '844', '839', '568', '216', '636', '877', '786', '832', '084', '536' |
| $\mathbf{h}_{post}^{19,12469}$, Subtraction |  | '344', '3', '444', '509', '4', '456', '445', '2', '333', '5' |
| $\mathbf{h}_{post}^{27,4900}$, Multiplication |  | '270', '409', '257', '454', '451', '570', '290', '287', '470', '439' |
| $\mathbf{h}_{post}^{20,11020}$, Division |  | '823', '983', '801', '731', '501', '751', '070', '663', '985', '713' |

