# OpenReview forum: "Arithmetic Without Algorithms: Language Models Solve Math with a Bag of Heuristics"
_ICLR.cc/2025/Conference — ICLR 2025 Poster_

### Official Review · Reviewer_fm6H · 2024-10-22

**Soundness:** 3
**Presentation:** 2
**Contribution:** 3
**Rating:** 6
**Confidence:** 3

**Summary:**

This paper characterizes which neurons correspond to LLM performance on arithmetic problems. First, it demonstrates that only a few attention heads in early layers are important, while MLP layers in later stages play a significant role. Through a more detailed analysis using methods like probing, the study identifies several crucial neurons in MLPs, discovering that certain operands can activate these neurons. Finally, the paper introduces the concept of "bag of heuristics" for these operand-activated neurons, claiming that this mechanism can account for how LLMs solve arithmetic problems.

**Strengths:**

The paper employs numerous experiments to validate its findings, and the discovery of neurons activated by specific operands is intriguing.

**Weaknesses:**

Despite applying various methods and experiments, the main contribution and conclusions drawn from the experiments are overclaimed and unclear. I summarize my primary concerns as follows:
1. The paper predominantly focuses on the MLPs in later layers, specifically on the last token ("=" sign), while neglecting the attention module in early layers. However, the representation of the last token in subsequent layers heavily depends on attention from early layers. Understanding how transformers utilize the attention mechanism for arithmetic problems is arguably more crucial than examining the function of subsequent layers.
2. While arithmetic equations have approximately 1,000,000 combinations, this study only samples 100 prompts for most of its results. This limited sample size introduces significant randomness and undermines the validity of the findings.
3. A critical issue is that this paper fails to provide a clear explanation of how LLMs solve arithmetic problems. For instance, we CANNOT explain how a transformer solves a problem like "21 + 17 =" in **more detail** than: "The LLM computes a representation for the last token based on the context in early layers, and then activates certain neurons to obtain the final result in subsequent layers."
4. The conclusions drawn from the experiments are often overclaimed and unclear. For example, in Section 4, the heuristic matching method is confusing. The results from this method don't seem to provide more useful information than the activation patching experiments. It appears that neurons from any model obtained by this method would likely show results similar to those in Figures 8 and 9.

**Questions:**

See weaknesses. Another question is about eq(3), can you explain h and v based on SwiGLU equation in https://paperswithcode.com/method/swiglu?

---

> ### Author Response · Authors · 2024-11-18
>
> Thank you for your feedback!
> We appreciate that you found our findings intriguing.
>
> To answer the reviewers comments and questions:
>
> \> **“The paper predominantly focuses on the MLPs in later layers, specifically on the last token ("=" sign), while neglecting the attention module in early layers. However, the representation of the last token in subsequent layers heavily depends on attention from early layers. Understanding how transformers utilize the attention mechanism for arithmetic problems is arguably more crucial than examining the function of subsequent layers.”**
>
> Previous research [3,4] has already investigated the effect of attention modules in early layers for arithmetic calculations, having shown their importance in information transfer to the final position.
>
> While we agree that understanding attention modules is crucial for examining circuit functionality, our work deliberately focuses on less-understood parts of the circuit.
> Our novel contributions focus on the role of later-layer MLPs, as the components that generate the correct answer (as pointed to in Section 2.2). We seek to answer the question “How do they generate the answer?” which is left unanswered by all previous work.
>
> Please note that we do not neglect attention heads. On the contrary, to provide a full image of the circuit functionality, we analyze the circuit attention heads in Appendix B.2. We demonstrate that these heads primarily project operand and operator information to the final position. In Appendix B.1, we further show that token embeddings moved by attention heads already contain important features initially promoted by MLP0 neurons, which are later identified by subsequent MLPs.
> Moreover, using faithfulness analysis (In Appendix A), we reveal that only a few heads are required to copy information from the token positions, further showing that their role (at least in Llama3-8B) is rather limited in the investigated context.
>
>
> \> **“While arithmetic equations have approximately 1,000,000 combinations, this study only samples 100 prompts for most of its results. This limited sample size introduces significant randomness and undermines the validity of the findings.”**
>
> To decrease randomness in our experiments, we have performed most of our experiments (circuit discovery and evaluation (Section 2.1 and Section 3.1), heuristic ablation experiments (Section 4.2, Section 5)) for several random seeds (50 seeds for Llama3-8B, Pythia-6.9B, GPT-J; 5 seeds for Llama3-70B, due to computational limits), leading to a larger prompts sample size. The results reported in the paper are the mean results across seeds.
>
> We have now added this detail to the updated revision in Appendix G (lines 998-1002), and thank the reviewer for pointing this out. The different seeds provide a more balanced sample across the data distribution, and produce low variance results.
>
> Due to the high computational requirements of activation patching (2 forward passes per batch per tested component), we are unable to repeat the circuit discovery process for ALL prompts. We thus use these representative samples, similarly to previous works [1,2,3, inter alia], and verify they lead to low-variance scores.

---

> ### Author Response · Authors · 2024-11-18
>
> \> **“A critical issue is that this paper fails to provide a clear explanation of how LLMs solve arithmetic problems. For instance, we CANNOT explain how a transformer solves a problem like "21 + 17 =" in more detail than: "The LLM computes a representation for the last token based on the context in early layers, and then activates certain neurons to obtain the final result in subsequent layers."”**
>
> We respectfully disagree with the reviewer on this comment. We believe that our claims do indeed explain the mechanism of arithmetic calculations in high detail, more than previously known in the literature [3,4].
>
> For this example prompt, “21+17=”, we can point to *specific* features of the input tokens op1 and op2 that are boosted by MLP0 (described in Appendix B.1), for example, the feature of “1mod10” written to op1 position and the feature of “<50” written to op2 position.
> In Llama3-8B, the collection of these features is copied by attention heads such as 16H21 that copies information from the op1 position directly to the last position, 15H13 that copies op2 features to the last position, and 2H2 that copies information regarding the operator itself to activate the correct neurons in later layers (described in Appendix B.2).
>
> Specifically, the mechanism we chose to focus on and explain is the later MLPs that promote the correct answer, which is relatively identical across models. In these layers, we are not only able to find the “certain neurons to obtain the final result”, but we further explain how they work. I.e. - we find several groups of specific neurons - some that promote the token “38” because it is higher than “21”, some that promote the token “38” because it is 0mod2 (and they activate because 21 and 17 are 1mod2), etc. The combination of the activations of these neurons, as we describe in Section 3 and provide evidence for in Section 4, is, in our claim, the mechanism behind the model’s successful completions of arithmetic prompts.
>
> Thus, our portrayal of the arithmetic calculation flow offers a more fine-grained perspective compared to both the reviewer’s description and the current understanding in the field [3,4]. Achieving such fine-grained insights across different tasks represents the main goal of mechanistic interpretability.
>
> \> **“The conclusions drawn from the experiments are often overclaimed and unclear. For example, in Section 4, the heuristic matching method is confusing. The results from this method don't seem to provide more useful information than the activation patching experiments.”**
>
> First, we aim to clarify - the point of the matching method and the following ablations in Section 4.1 and 4.2 is completely different from activation patching. In the patching experiments, we aim to search for components with high effect on the prompt computation process, while here we aim to label some of these components (the late-layer MLP neurons) into pre-defined roles (heuristic types).
> In Section 4.1 and 4.2 we present several experiments to indicate that the labels we provide are indeed meaningful: we show that for each arithmetic prompt, we can identify (using the “labels” we provide) a collection of individual neurons that implement unrelated heuristics, and only when the contribution of all of these neurons is combined does the model generate the correct answer.
>
> Second, if there are any remaining unclear conclusions, we would be happy to provide more information.
>
> \> **“It appears that neurons from any model obtained by this method would likely show results similar to those in Figures 8 and 9.”**
>
> This is inaccurate. If you mean that ablating any random group of MLP neurons will harm the model’s arithmetic performance similarly to a group of targeted “heuristic neurons” found by our algorithm, this is *exactly* what we measure as the baseline in Figure 8 - we show that ablating a random set of neurons affects prompts much less than ablating the heuristic neurons associated with these prompts (when these are labeled by our algorithm).
>
> \> **“Another question is about eq(3), can you explain h and v based on SwiGLU equation in paperswithcode.com/method/swiglu?”**
>
> If we observe the definition of SwiGLU from its defining paper as:
>
> $FFNSwiGLU(x, W, V, W_2) = (Swish(xW ) ⊗ xV) W_2$
>
> where $W, V, W_2$ are weight matrices, than the vector $h_{post} = (Swish(xW ) ⊗ xV)$ will be the equivalent of “Key activations”, which we define in the paper as the vector of neurons. Our definition of $v_{out}$ will become row vectors in $W_2$ in the case of a SwiGLU activation function.
>
>
>
> We hope our answers and updated revision are satisfactory. Please let us know if you have any other questions or concerns.

---

> ### Author Response · Authors · 2024-11-18
>
> [1] Hanna et. al, “How does GPT-2 compute greater-than?: Interpreting mathematical abilities in a pre-trained language model”
>
> [2] Wang et. al, “Interpretability in the wild: A circuit for indirect object identification in GPT-2 small”
>
> [3] Stolfo et. al, “A Mechanistic Interpretation of Arithmetic Reasoning in Language Models using Causal Mediation Analysis”
>
> [4] Zhang et. al, “Interpreting and improving large language models in arithmetic calculation.”

---

> > ### Comment · Reviewer_fm6H · 2024-11-21
> >
> > I thank the authors for their responses and additional explanation. I will increase my rating from 5 to 6 as I support accepting this work.

---

### Official Review · Reviewer_jXdJ · 2024-11-04

**Soundness:** 3
**Presentation:** 2
**Contribution:** 3
**Rating:** 6
**Confidence:** 4

**Summary:**

This paper investigates how LLMs solve arithmetic tasks, questioning whether they rely on robust algorithms or if they instead depend on memorization or simple heuristics. Using methods like activation patching and linear probing, the authors identify a subset of neurons responsible for arithmetic tasks in LLMs, which they term the “arithmetic circuit.” The study reveals that LLMs rely on a "bag of heuristics" formed by a small subset of individual neurons, each triggered based on specific input patterns. Through various experiments and ablations, the paper demonstrates that this heuristic-based mechanism is the primary method for handling arithmetic prompts, even from early stages of model training. The authors conclude that LLMs’ arithmetic capabilities do not stem from robust algorithms or mere memorization, but from a combination of these lightweight heuristics.

**Strengths:**

1. The paper makes a valuable contribution by providing a mechanistic interpretation of how LLMs handle arithmetic tasks, adding a nuanced understanding of model behavior beyond existing studies on memorization and algorithmic capability. By identifying individual neurons and describing them as "heuristic carriers," this study offers a novel perspective on LLM internal operations.
2. The authors clearly articulate their methodology and conclusions. The visualizations, especially those illustrating neuron activation patterns, support comprehension and substantiate the paper’s core claims about the heuristic-based arithmetic mechanism.
3. The paper employs causal analysis techniques to dissect the arithmetic circuit in LLMs and validate its findings with multiple ablation experiments. The robustness of this methodological approach makes the proposed hypothesis and claims convincing.

**Weaknesses:**

1. While the study primarily focuses on arithmetic tasks, it does not fully explore whether this heuristic-driven mechanism applies to other types of reasoning. Future work extending the approach to different reasoning domains would help determine the generalizability of these findings.
2. Classifying neurons into heuristic types may oversimplify the actual complexity of neuron interactions in LLMs. Additionally, the manual classification of heuristic types could introduce subjective bias.

**Questions:**

1. How would the proposed heuristic-based mechanism scale with model size or complexity? Does the number or type of heuristics vary significantly with larger models?
2. Would small models trained from scratch on arithmetic tasks also rely on a "bag of heuristics"? Or would they have a different mechanism to learn arithmetic? One suggestion is to train a small model (like using a NanoGPT framework) from scratch and apply the same analysis.
3. Are there any insights into how these heuristics interact or overlap in other reasoning contexts, such as multi-step reasoning tasks? Does the mechanism generalize across tasks?

---

> ### Author Response · Authors · 2024-11-18
>
> Thank you for your feedback!
> We appreciate that you found our contributions to be novel and valuable to understanding LLMs behaviors. We also value that you found our methodologies robust and clear, and were convinced by our claims and conclusions.
>
> As per your comment on applying our conclusions to more general reasoning tasks, and per reviewer H4kU’s suggestion, we have added this as an additional discussion point in our limitations (lines 536-539).
>
> To answer the reviewer’s remaining questions and comments:
>
> \> **“Classifying neurons into heuristic types may oversimplify the actual complexity of neuron interactions in LLMs.“**
>
> We believe that because we work in a limited data distribution of arithmetic prompts, it makes sense to ignore other complex roles of these neurons (for prompts in other data distributions), which most likely exist (due to neuron superposition [1]). We don’t claim these neurons are ONLY arithmetic neurons, but do claim that when processing arithmetic prompts, these neurons are paramount in identifying numerical patterns and promoting relevant arithmetic answers.
>
> \> **”Additionally, the manual classification of heuristic types could introduce subjective bias.”**
>
> First, we would like to correct the reviewer and say that our classification process is automated (the algorithm is described in Section 4.1 and Appendix F), and this, in our opinion, represents progress compared to prior mechanistic work on understanding model component roles, which was completely manual (for example, [4]).
> The manual part in the process is the definition of heuristic types, and this part indeed introduces human bias, a limitation we discuss in our limitations section. We hope further work in the field of automated interpretability might also allow decreasing human bias when defining labels for roles of model components, and will allow the process of understanding neurons to be automated even further than we did in the paper.
>
>
> \> **”Would small models trained from scratch on arithmetic tasks also rely on a "bag of heuristics"? Or would they have a different mechanism to learn arithmetic? One suggestion is to train a small model (like using a NanoGPT framework) from scratch and apply the same analysis.”**
>
> Previous works that we cite in our related work section performed an analysis similar to your proposal [2, 3]. They trained toy models from scratch on a VERY limited data distribution (i.e. only modular addition). In their different setting they show that toy models can learn features similar to those employed in larger pre-trained models (i.e. Fourier Features to represent numbers). However, the utilization of these features seems to be completely different in toy models when compared to large pre-trained models. The authors of [2,3] found very specialized algorithms that use the numerical features (which they term the “Clock” algorithm or the “Pizza” algorithm [3]) - algorithms that are completely different from the “bag of heuristics” we present in larger models. We believe the distinction of the two mechanisms between large models (shown by our work) and toy models (shown in [2,3]), and what causes this distinction, offers a very interesting case study for further work.
>
> \> **“How would the proposed heuristic-based mechanism scale with model size or complexity? Does the number or type of heuristics vary significantly with larger models?”**
>
> We answer this question in the paper in two main scales - roughly 8B (we use Pythia-6.9B, GPT-J and Llama3-8b) and 70B (Llama3-70B). We show our findings apply across these scales.
> We found that the similar heuristic-based mechanisms scale and exist also in the larger model. As we show in Appendix I, Figure 16c, Llama3-70B follows the same trends of Llama3-8B (requiring a very low amount, roughly 150~, neurons per MLP in the middle- and late-layer MLPs; requiring a very small amount of specialized attention heads).
> Due to hardware availability limitations, we couldn’t investigate Llama3-70B further than we did, and it is possible that further manual investigation of its heuristic neurons / an automated process to classify them can lead to additional heuristic types that only appear in higher scales (although we doubt it due to the similar ablation results in Llama3-70b that we show in Figure 8b).

---

> ### Author Response · Authors · 2024-11-18
>
> \> **“Are there any insights into how these heuristics interact or overlap in other reasoning contexts, such as multi-step reasoning tasks? Does the mechanism generalize across tasks?”**
>
> This is a good question, which we hope future work can explore. It is possible that similar “bags of heuristics” exist for other textual tasks, however we feel it is premature of us to claim this, as our work focused on arithmetic reasoning.
>
> We hope our answers and updated revision are satisfactory. Please let us know if you have any other questions or concerns.
>
>
> [1] Elhage et. al, “Toy models of superposition”
>
> [2] Nanda et. al, “Progress measures for grokking via mechanistic interpretability”
>
> [3] Zhong et. al, “The clock and the pizza: two stories in mechanistic explanation of neural networks”
>
> [4] Wang et. al, “Interpretability in the wild: A circuit for indirect object identification in GPT-2 small”

---

> > ### Comment · Reviewer_jXdJ · 2024-12-02
> > **Official Comment by Reviewer  jXdJ**
> >
> > Thank you for answering my questions and providing clarifications. I would like to maintain my scores.

---

### Official Review · Reviewer_H4kU · 2024-11-04

**Soundness:** 4
**Presentation:** 3
**Contribution:** 3
**Rating:** 8
**Confidence:** 3

**Summary:**

It is unclear how large language models solve reasoning tasks. For example, do they simply memorize the solution or learn generalizable reasoning processes. This papers demonstrate language models use a set of heuristics to solve arithmetic (a task where we expect a successful learner to learn generalizable algorithms). The paper characterizes the different types of heuristics being implemented as how heuristics are developed during training.

**Strengths:**

- For the most part the paper was well-written and well motivated. At any given point in the paper, it was easy to understand (1) what the hypothesis being tested is, (2) why the authors are testing this hypothesis, (3) the experimental setup, and (4) the results.
- The experiments conducted by the authors are thorough and well-support the claims made throughout the paper.
- The topic of the paper is timely and illuminates an interesting aspect of arithmetic reasoning in language models that has not been explored before. I think this will add value both to those working in mechanistic interpretability (because of the clean experimental designs) as well as those working in NLP in general (as it provides an important case study).
- The paper's results are replicated across models of different scales as well.

**Weaknesses:**

- The title of the paper is slightly misleading: “Models Sovle Math with a Bag of Heuristics” in reality the authors only examine arithmetic where both operands have less than three digits. Maybe for greater number of digits, these heuristics are somehow algorithmically combined? This question is left open in the paper (which is fine, I think the paper is a good contribution as is), but the title does not reflect this.
    - In the very least, I think this point should be mentioned briefly in the limitations section.
- There are some experimental details which are shoved under the rug which I cannot find the appendices (see below). I think adding (probably in the appendices) these would greatly help the reproducibility of the results.

**Questions:**

- Eq. 2, pg. 4, why are the authors measuring faithfulness by averaging the percentage change in logits? It seems that a more principled approach would be checking how much the correctness of the model (measured through accuracy) decreases as a result of mean ablation. Did the authors ablate over this design variable?
- In Fig. 7, I am confused by what the x-axis is here. It would make sense that it the accumulation of heuristics, but that is not what the axis label implies. Could the authors elaborate on this a bit more?
- In line 321, the authors use $t=0.6$ as the intersection threshold, why was this hyperparameter specifically chosen, was it based on the alignment with manual insepction? I do not see ablations over this parameter in the appendices.
- In lines 408-410, “incorrect prompts have more heuristic neurons associated with them than correct prompts” how are the authors measuring this involvement? Is this through the same sort of patching that was done before, but in this case how are the authors patching the neurons for a baseline incorrect response?
- It has been shown in some recent works that logic lens in very often unreliable see [1]. Have the authors tried other hidden state decoding methods like [1,2,3]? I wonder if this could be used to better classify some of the heuristics that did not fall into any of the identified categories.
- In lines 347-355, the authors perform knockout experiments. I am wondering if we knockout say a neuron associated with the heuristic $\textrm{op}_1 \equiv 0\mod2$ that significantly contribues sto the output, we not only see a decrease in performance, but also a *decrease in performance on inputs specifically where the first operand is even?* This would provide even stronger evidence for not only the importance of the heuristics, but also the claimed functionality of each of them.
    - Is this what Fig. 7 is trying to show, specifically, the importance of the “associated prompts?”

[1] Belrose, Nora, et al. "Eliciting latent predictions from transformers with the tuned lens." *arXiv preprint arXiv:2303.08112* (2023).

[2] Pal, Koyena, et al. "Future lens: Anticipating subsequent tokens from a single hidden state." *arXiv preprint arXiv:2311.04897* (2023).

[3] Ghandeharioun, Asma, et al. "Patchscope: A unifying framework for inspecting hidden representations of language models." *arXiv preprint arXiv:2401.06102* (2024).

---

> ### Author Response · Authors · 2024-11-18
>
> Thank you for your feedback!
> We very much appreciate that you found our work novel and mentioned its importance to mechanistic interpretability and NLP in general. We also appreciate that you found our writing and experiments as clear and well motivated.
>
> We have applied your proposal regarding explicitly mentioning the setting as an additional limitation in the updated revision (lines 536-539).
>
> We would like to answer to the reviewer’s remaining questions and comment about experimental details:
>
> \> **“In lines 347-355, the authors perform knockout experiments. I am wondering if we knockout say a neuron associated with the heuristic op1≡0mod2 that significantly contributes to the output, we not only see a decrease in performance, but also a decrease in performance on inputs specifically where the first operand is even? This would provide even stronger evidence for not only the importance of the heuristics, but also the claimed functionality of each of them.Is this what Fig. 7 is trying to show, specifically, the importance of the “associated prompts?””**
>
> Yes! This is exactly the experiment shown in Figure 7! We thank the reviewer for proposing the ablation and acknowledging it provides stronger evidence of our claims.
>
> \> **“In Fig. 7, I am confused by what the x-axis is here. It would make sense that it the accumulation of heuristics, but that is not what the axis label implies. Could the authors elaborate on this a bit more?”**
>
> The x axis values represent indices of heuristics. For example, when observing the addition operator, we identify (in Section 4.1) roughly 70 heuristic types into which the neurons classify into. We perform the first ablation experiment described in Section 4.2 on each heuristic index *i*, to get two measures, marked in blue and yellow in Figure 7 at x=i.
> The reason for the almost monotonic increase is because the heuristics are sorted based on the difference between the baseline accuracy (an almost vertical line around 0.95~, not presented in the graph for simplicity), and the ablated accuracy of associated prompts (the blue line). Thus the heuristics where the ablation of neurons is the most successful appear first.
>
>
> \> **“In line 321, the authors use t=0.6 as the intersection threshold, why was this hyperparameter specifically chosen, was it based on the alignment with manual inspection? I do not see ablations over this parameter in the appendices.”**
>
> We present the reason to choose this value in Appendix G, under "heuristic classification". The threshold value affects the precision and recall of the classification. When increasing the threshold, the recall will decrease (more FN, since we miss good neuron classifications into heuristics). When decreasing the threshold, the precision decreases since there will be more FP cases. A good threshold value should balance between these two.
> The lack of ground truth labels for the neurons meant we needed to manually check at least some of the classifications to get a sense for the precision and recall, meaning we couldn't perform a complete ablation across the possible values of t. Thus, we experimented with several values, in each manually checking some of the classifications (only some because we have over 3000 neurons), and chose 0.6 as a manual pareto-”optimal” value that balances between precision and recall.
>
> \> **“It has been shown in some recent works that logic lens is very often unreliable see [1]. Have the authors tried other hidden state decoding methods like [1,2,3]? I wonder if this could be used to better classify some of the heuristics that did not fall into any of the identified categories.”**
>
> We haven’t tried any of these methods, and we thank the reviewer for pointing them out! Regarding Future Lens [2], it seems to apply mostly when attempting to extract information for later position activations, while we use logit lens on weight vectors (row vectors of the second linear layer in MLPs). Tuned lens [1] (and its generalization in Patchscopes [3]) also apply to activations and not weights, and require a training process for a “translator layer” to overcome the limitations of logit lens by “imitating” the rest of the computation the activations pass in the transformer. It is unclear how to convert this training process to work on weight vectors (due to the limited amount of them). However, if this can be solved, it might prove to be an interesting approach to better understand the logits of tokens embedded in middle-layer MLP weights and not just activations.

---

> ### Author Response · Authors · 2024-11-18
>
> \> **“Eq. 2, pg. 4, why are the authors measuring faithfulness by averaging the percentage change in logits? It seems that a more principled approach would be checking how much the correctness of the model (measured through accuracy) decreases as a result of mean ablation. Did the authors ablate over this design variable?”**
>
> We did! The reason we didn’t use accuracy is because it is not continuous:
> When trying to compare a circuit C that contains a set of components $S$ to a circuit $C^*$ that contains the set $S \cup \{c\}$ (where $c$ is some attention head (as seen in Appendix A) / MLP neuron (As seen in Figure 4b)), we want to get a fine-grained sense for the importance of the component $c$.
> If you think about the (rather common) case where mean-ablating all components out of $S$ leads to a logit $l_1$ for the correct label, and $l_2$ for an incorrect chosen label, such that $l_1 <<  l_2$, it is possible that adding $c$ to the circuit leads to an increase in $l_1$, marking its importance in evaluation. But if it doesn’t cause $l_1$ to increase enough such that $l_1 \gt l_2$, the evaluation using accuracy won’t show it is important at all! On the other hand, if we evaluate using the logit $l_1$ normalized by the maximal logit, we will succeed in measuring this increase. Thus, we use the normalized logit measure for a “smoother” evaluation metric over circuit component count.
> Following the reviewers request, we have added this discussion point as an implementation detail in Appendix G (lines 1016-1026).
>
> \> **“In lines 408-410, “incorrect prompts have more heuristic neurons associated with them than correct prompts” how are the authors measuring this involvement? Is this through the same sort of patching that was done before, but in this case how are the authors patching the neurons for a baseline incorrect response?”**
>
> No, we measure the “involvement”(association, as we term it in the paper) of a neuron with a prompt, by observing the properties of the prompt itself. For example, for the prompt “1+331=” we can say that it is associated with the heuristic “op1 in range(0, 10)” (and many more). Thus, if there are any neurons that implement this heuristic, we assume they should somehow affect the completion of this prompt, and we count them. In Section 4.3, for each prompt (correct / incorrect) we count neurons that implement any of the heuristics that are associated with the prompt (lines 405-406) to check our hypothesis that correct prompts merely have more neurons associated with them.
>
> We hope our answers and updated revision are satisfactory. Please let us know if you have any other questions or concerns.

---

> ### Comment · Reviewer_H4kU · 2024-11-26
>
> Thanks to the authors for such a detailed response! All of my questions have been answered. I believe this is a strong paper and will recommend the paper be accepted.

---

### Official Review · Reviewer_gBdd · 2024-11-04

**Soundness:** 3
**Presentation:** 2
**Contribution:** 3
**Rating:** 6
**Confidence:** 2

**Summary:**

This paper explores the reasoning mechanisms behind arithmetic calculations in large language models (LLMs). The authors use causal analysis to identify an "arithmetic circuit" responsible for these calculations, finding that LLMs apply a set of heuristic-based neurons rather than true algorithmic methods or pure memorization. Each neuron in this circuit encodes simple, pattern-based heuristics that activate for specific operand patterns and contribute to the model's ability to solve arithmetic prompts. The paper provides a comprehensive examination of these heuristics and their emergence over training time, shedding light on the limitations and generalization abilities of LLMs in mathematical reasoning.

**Strengths:**

This paper applies causal analysis and activation patching experiments to identify individual neuron behaviors.

The authors' step-by-step examination, from identifying neurons and classifying them into heuristic types to analyzing their evolution over training time, offers a thorough perspective on LLMs' arithmetic reasoning mechanisms.

The paper may facilitate future studies on generalization abilities.

**Weaknesses:**

This paper focuses on arithmetic calculation but mentions general reasoning at the beginning. I feel that this could be a overclaim because simple one-step calculation is far away from reasoning.

The authors get all conclusions from pre-trained checkpoints. But I am thinking that all models need fine-tuning before usage, it is better to have similar analysis on arithmetically or generally fine-tuned LLMs.

The focus is primarily on arithmetic reasoning, which may not fully generalize to other complex reasoning tasks. It is interesting to see the model behavior when arithmetic calculation is embedded in texts.

**Questions:**

The classification of neurons into heuristic types was manually conducted. Do you think if it is possible to develop an automated or semi-supervised system to reduce human bias?

You mention that the "bag of heuristics" mechanism evolves gradually during training. Have you observed whether certain heuristics are reinforced at specific training stages?

Your analysis reveals cases where the model’s heuristics do not generalize to certain arithmetic prompts. Could you provide more insights into these failure cases? For instance, are there specific numerical ranges or types of prompts where the model tends to fail more frequently?

---

> ### Author Response · Authors · 2024-11-18
>
> Thank you for your feedback!
> We appreciate that you found our work to be thorough and give a high resolution understanding of LLMs’ arithmetic mechanisms, as well as point towards further studies on understanding generalization in arithmetics in LLMs.
>
> We have taken into account the comment regarding “mentioning general reasoning as an overclaim”. We don’t aim to explain the mechanisms behind all types of reasoning, but hope our findings in the basic cases can lead to further work on this problem.
> Per reviewer H4kU’s suggestion, we added this as a discussion point in our limitations (lines 536-539).
>
> To answer the rest of the reviewer’s comments and questions:
>
> \> **"The authors get all conclusions from pre-trained checkpoints. But I am thinking that all models need fine-tuning before usage, it is better to have similar analysis on arithmetically or generally fine-tuned LLMs."**
>
> This is a very interesting comment. When defining our experimental settings, we chose to use only pre-trained models (without fine-tuning) due to 2 main reasons:
> First, we tried to keep the model as “clean” as possible - fine-tuning might lead to a number of effects depending on the type of fine-tuning and training data, and we tried to keep our conclusions as general as possible.
> Second, in [1] it was shown that fine-tuning does not replace the mechanisms learned by LLMs, but merely enhances them. This indicated that fine-tuning won’t change the underlying algorithm, and that our analysis will yield the same results on a pre-trained model and its fine-tuned version (for example, Llama and Goat). This hypothesis is strengthened by results from [2], who located the same arithmetic circuit structure in a pre-trained and fine-tuned LLM, making us think that fine-tuning on arithmetic data will merely strengthen the bag of heuristics we present in this work.
>
> \> **"The classification of neurons into heuristic types was manually conducted. Do you think if it is possible to develop an automated or semi-supervised system to reduce human bias?"**
>
> Yes! Developing an automated approach for component classification is a very promising direction. We have made some attempts in this direction. Following some attempts done by previous work [3], we tried to use existing LLMs (ChatGPT, Claude) to classify high-activating prompt sequences (the same ones seen to fire in the patterns in Figure 1), but failed to get reliable results. We therefore hope that further work in this field can help research questions like ours become more automated and less human biased, as we also discuss in the limitations section.
>
> \> **"You mention that the "bag of heuristics" mechanism evolves gradually during training. Have you observed whether certain heuristics are reinforced at specific training stages?"**
>
> Yes - In section 5, we noticed that all heuristic types across all four arithmetic operators interestingly had the same gradual increase. Thus, we chose to present the average line across heuristic types and operators in Figure 10a. A different result might’ve been more interesting to look into (why do some heuristics develop before others).
> Following this comment, we have added standard deviation markings in Figure 10a, to show that the deviation from the mean gradual increase is relatively low. We thank the reviewer for pointing this out.
>
> \> **"The focus is primarily on arithmetic reasoning, which may not fully generalize to other complex reasoning tasks. It is interesting to see the model behavior when arithmetic calculation is embedded in texts."**
>
> Regarding "may not fully generalize to other complex reasoning tasks", we have added this as a discussion point in our limitations section, per reviewer's H4kU's suggestion.
>
> Regarding “prompts where arithmetic calculation is embedded in texts”, previous work [2] has shown that there is a similar structure of the arithmetic circuit for several templates of arithmetic operators, including textual prompts (e.g. “How much is x plus y?”), even if they differ in positions. Thus, we believe our findings *do* generalize to such templates, while we focused on analyzing numerical templates in the scope of this work.
> Looking further into more complex reasoning tasks (or more complex arithmetic settings) is a very interesting direction for future work, and we hope our work acts as a stepping stone in that direction.

---

> ### Author Response · Authors · 2024-11-18
>
> \> **"Your analysis reveals cases where the model’s heuristics do not generalize to certain arithmetic prompts. Could you provide more insights into these failure cases? For instance, are there specific numerical ranges or types of prompts where the model tends to fail more frequently?"**
>
> We looked into these failure cases, and unfortunately found no shared property of the prompts that fail to generalize (other than some less interesting findings such as "the model fails less when the operands are single-digit"). This finding led us to defining the experiments in Section 4.3 - we ask “if there are no shared properties of the failed heuristics, what makes the model fail to generalize on them?”, and we found that the failed prompts are not special - the heuristics associated with them merely promote the correct answer logit slightly less than other prompts.
>
> We hope our answers and updated revision are satisfactory. Please let us know if you have any other questions or concerns.
>
> [1] Prakash et. al, “Fine-tuning enhances existing mechanisms: a case study on entity tracking”
>
> [2] Stolfo et. al, “A mechanistic interpretation of arithmetic reasoning in language models using causal mediation analysis”.
>
> [3] Bills et. al, “Language models can explain neurons in language models”

---

### Official Review · Reviewer_Tumw · 2024-11-05

**Soundness:** 4
**Presentation:** 3
**Contribution:** 3
**Rating:** 8
**Confidence:** 4

**Summary:**

The paper uses causal analysis to understand how (small scale) LLMs solve arithmetic. They identify that the model's output really depends on only a small subset of its parameters which they call a circuit and use this to explain its behavior. They find that each neuron in this circuit implements some form of a heuristic and the model's overall behavior can be explained as a collection of these simple heuristics. They run extensive experiments and analysis to validate their claims and show that somewhat surprisingly, the model is neither relying purely on memorization or learning a robust generalizable algorithm for addition or multiplication.

**Strengths:**

1. Arithmetic tasks are a particularly useful test bed to understand the learning mechanism of LLMs. However, it has been an open question as to what algorithm the model is truly learning. This work makes an important step in this direction, particularly since it applies to the newest edition of LLMs.
2. The bag of heuristics finding explains the lack of length generalization in most LLMs on arithmetic tasks.
3. The experiments and analysis are extensive and the authors make an effort to validate all of their claims sufficiently.
4. The paper is well written with the ideas explained clearly with good intuition.

**Weaknesses:**

1. A lot of the analysis assumes that the tokenizers all tokenizers numbers as a single token up to some limit. While I found this interestingly to be true for llama3 and pythia, I don't think this is always true. In fact, the llama2 tokenizer itself was different in that it tokenized each digit individually. I would like to see if these findings are tokenizer specific.
2. The sampling mechanism of activation patching seems to be of fairly "high-variance". Sampling a "random counterfactual prompt" from the universe of prompts seems to me like it would lead to widely different results. Is this true?
3. The difference between the total logit contribution of the heuristc neurons to the correct answers vs the incorrect answers is fairly small. And thus it is plausible that this is not the only (or even main) failure mode of the bag of heuristics.
4. The experiments on tracking development across training steps is on a model with a very different tokenizer and much worse performance than Llama3-8B. I understand that this is necessary since llama3-8b's training data is not public, however it does mean the results are slightly harder to compare.

**Questions:**

1. In Figure 3, is the position of Operator 1, Operator 2 a range? Or are the inputs only up to the point where it is a single token?
2. Am I right in understanding that once the circuit is identified, the network can be replaced with a highly sparse version of the same network? (Say 98.5% sparse where the fixed components can be replaced by biases?)
3. Does the bag heuristics finding imply that length generalization for these models is impossible?

---

> ### Author Response · Authors · 2024-11-18
>
> Thank you for the feedback!
> We appreciate that you found our paper as an important step towards understanding model-implemented algorithms, and mentioned the extensiveness of our experimental analysis.
>
> We would like to answer the reviewer’s questions and comments:
>
> \> **“Am I right in understanding that once the circuit is identified, the network can be replaced with a highly sparse version of the same network? (Say 98.5% sparse where the fixed components can be replaced by biases?)”**
>
> Yes! If we replace the activations of each component with a pre-computed mean activation, and only inference the sparse arithmetic circuit, most of the performance (roughly 96%, see lines 173 and 244) of the model for all arithmetic prompts can be recreated. This is exactly what we measure in our faithfulness evaluation experiments in sections 2 and 3.
>
> \> **“Does the bag heuristics finding imply that length generalization for these models is impossible?”**
>
> Not necessarily. Further improvements, both to the training process and the inference process (for example, Chain of Thought prompting) might allow dividing more complex arithmetic tasks to simpler tasks that allow using the heuristics we found. However, without such improvements, we believe that the heuristics explain the failure of such models to generalize to harder prompts. For example, in the slightly harder setting when op1,op2 \in [1000, 3000], and allowing the model to answer in many tokens, the accuracy of Llama3-8B drops to around 0.1 for addition and subtraction, and near zero for multiplication and division).
>
> \> **“In Figure 3, is the position of Operator 1, Operator 2 a range? Or are the inputs only up to the point where it is a single token?”**
>
> Single token. As stated in Section 2, we use templates with single-token operands (up until op=1000 in Llama3-8B) for simplicity.
>
> \> **“A lot of the analysis assumes that the tokenizers all tokenize numbers as a single token up to some limit.”**
>
> This is true, and we acknowledge this in our limitations section. Our analysis applies to most models, which tokenize digits in groups, and might lead to different conclusions on models (e.g. Llama2, Mistral) that tokenize numbers per-digit.
> However, previous work [1] has located an arithmetic circuit in such a setting on Llama2-7B, and have shown that in single-digit tokenization, similar components are important - namely the later MLPs, as well as some subset of attention heads. This is an initial evidence that the underlying algorithm shares functionality with the bag of heuristics mechanism we found in our work.
>
> \> **“The sampling mechanism of activation patching seems to be of fairly high-variance...is this true?”**
>
> While this is true, and the choice of counterfactual affects the components that are pointed to as being important, we found that a varied choice of counterfactuals across prompts (i.e. some differing in op1 values, some in op2 values, and some in the operator) is enough to give a good average. This is strengthened by repeating the patching experiments with different random seeds and larger prompt sets (Added to Appendix G, lines 998-1002), that still point to the same important components.
>
>
> We hope our answers and updated revision are satisfactory. Please let us know if you have any other questions or concerns.
>
>
>
> [1] Zhang et. al, "Interpreting and improving large language models in arithmetic calculation"

---

> > ### Comment · Reviewer_Tumw · 2024-11-24
> > **Thank you for your responses**
> >
> > Thank you for answering all of my questions. I will retain my score and recommend that the paper be accepted.

---

### Author Response · Authors · 2024-11-25

We appreciate the feedback provided by all the reviewers on our work and the time they have put in. We are thankful that the paper has received positive feedback. As we approach the end of the discussion period, we welcome any remaining questions and would appreciate confirmation from the reviewers on whether our responses have sufficiently addressed their concerns.

---

### Meta-Review · Area_Chair_8c6j · 2024-12-22

**Metareview:**

This paper reveals that a small subset of neurons governs a LLM's arithmetic performance. These neurons implement heuristic rules. The authors show that arithmetic behavior stems neither from simple memorization nor a generalized algorithm, but from a collection of heuristics implemented by them. Their causal analysis, ablation studies, and explanation of neuron activation patterns strongly support this claim.

In the rebuttal, the authors clarified many questions raised by the reviewers (different tokenization strategies and sampling methods, how these heuristics might scale to larger models or different tasks.) All the reviewers found the paper very interesting, and the authors' responses were satisfactory, maintaining high scores.

**Additional Comments On Reviewer Discussion:**

see above

---

### Decision · Program_Chairs · 2025-01-22

Accept (Poster)